# L-Diffusion: Laplace Diffusion for Efficient Pathology Image Segmentation

**Weihan Li** [1 2] **Linyun Zhou** [1 2] **Jian Yang** [1 2] **Shengxuming Zhang** [1 2] **Xiangtong Du** [3] **Xiuming Zhang\*** [4]
**Jing Zhang** [4] **Chaoqing Xu\*** [5] **Mingli Song** [1 2 6] **Zunlei Feng\*** [1 2 6]

## Abstract

Pathology image segmentation plays a pivotal role in artificial digital pathology diagnosis and treatment. Existing approaches to pathology image segmentation are hindered by labor-intensive annotation processes and limited accuracy in tail-class identification, primarily due to the long-tail distribution inherent in gigapixel pathology images. In this work, we introduce the Laplace Diffusion Model, referred to as L-Diffusion, an innovative framework tailored for efficient pathology image segmentation. L-Diffusion utilizes multiple Laplace distributions, as opposed to Gaussian distributions, to model distinct components—a methodology supported by theoretical analysis that significantly enhances the decomposition of features within the feature space. A sequence of feature maps is initially generated through a series of diffusion steps. Following this, contrastive learning is employed to refine the pixel-wise vectors derived from the feature map sequence. By utilizing these highly discriminative pixel-wise vectors, the segmentation module achieves a harmonious balance of precision and robustness with remarkable efficiency. Extensive experimental evaluations demonstrate that L-Diffusion attains improvements of up to 7.16%, 26.74%, 16.52%, and 3.55% on tissue segmentation datasets, and 20.09%, 10.67%, 14.42%, and 10.41% on cell segmentation datasets, as quantified by DICE, MPA, mIoU, and FwIoU metrics. The source codes are available at https://github.com/Lweihan/LDiffusion.

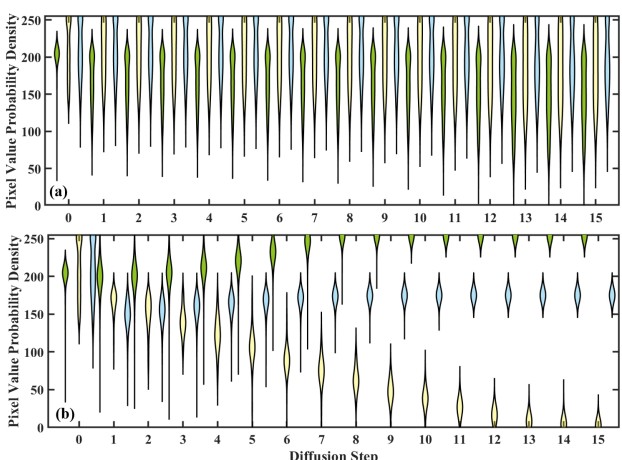

*Figure 1.* A comparative analysis of latent feature distributions between standard diffusion (a) and L-Diffusion (b). The latent feature distributions for individual components across various diffusion steps are denoted by violin plots in green, yellow, and cyan color.

[1]State Key Laboratory of Blockchain and Data Security, Zhejiang University [2]School of Software Technology, Zhejiang University [3]School of Medical Imaging, Xuzhou Medical University [4]The First Affiliated Hospital, College of Medicine, Zhejiang University [5]School of Computer and Computing Science, Hangzhou City University [6]Hangzhou High-Tech Zone (Binjiang) Institute of Blockchain and Data Security. Correspondence to: Zunlei Feng <zunleifeng@zju.edu.cn>, Xiuming Zhang <xm_zhang@zju.edu.cn>.

*Proceedings of the 42$^{th}$ International Conference on Machine Learning*, PMLR 267, 2025. Copyright 2025 by the author(s).

## 1. Introduction

Pathology images encompass a wealth of multi-level tissue and cellular features, regarded as the "gold standard" for cancer diagnosis. Pathologists meticulously examine and analyze the intricate relationships between diverse tumors and the characteristics of their components, including global lesions, tissue proportions and distributions, as well as cellular morphology (McGenity et al., 2024).

Beyond the tumor itself, the tumor microenvironment comprises a complex array of elements, such as surrounding blood vessels, tissues, and cells, all of which play pivotal roles in tumorigenesis, progression, metastasis, and treatment response. Consequently, numerous researchers have directed their efforts toward the segmentation of multi-level features within pathology images, aiming to enhance tumor diagnosis and microenvironment analysis. This, in turn, facilitates a deeper understanding of the mechanisms underlying tumorigenesis, progression, and metastasis (Zhao et al., 2021; Ye et al., 2023; Hosseini et al., 2024).

Consequently, researchers have devised advanced pathology image segmentation techniques to streamline the analysis of such intricate images. Prominent segmentation models,

including U-Net (Ronneberger et al., 2015), DeepLab (L.-C. Chen et al., 2017) and Transformer (Atabansi et al., 2023), have demonstrated exceptional performance across a spectrum of medical image classification tasks. Prior studies have also successfully integrated those deep learning models into pathology tissue segmentation (Ye et al., 2023) and cell segmentation (Hayakawa et al., 2021).

Pathology images exhibit three distinct characteristics: gigapixel resolution, a broad spectrum of scales, and a long-tail distribution. The annotation of gigapixel pathology images demands considerable time and effort from pathologists (Davri et al., 2022; Mahmood et al., 2023). Additionally, diverse pathological features often manifest at varying scales within the same image, necessitating models capable of capturing multi-scale information (Tarekegn et al., 2024). Furthermore, the imbalance in tissue type distributions complicates the extraction of features from tail categories (Ding et al., 2022), thereby intensifying the challenges of model training. In summary, current pathology image segmentation tasks grapple with labor-intensive annotation processes or limited accuracy in identifying tail samples.

Motivated by the diffusion model's exceptional ability to model latent distributions, we introduce the Laplace Diffusion Model, referred to as L-Diffusion, for efficient pathology image segmentation in this work. In lieu of the standard Gaussian distribution, we employ multiple Laplace distributions to model distinct components. Theoretical analysis demonstrates that the Laplace distribution is advantageous for broadening distribution disparities.

For the input original image, L-Diffusion initially transforms it into a sequence of intermediate reconstructed feature maps through $T$ diffusion steps. Each reconstructed feature map is subsequently converted into a grayscale image, thereby generating a new sequence of feature maps. Along the diffusion step axis, each pixel corresponds to a vector $v \in \mathbb{R}^{1 \times T}$, referred to as the pixel latent vector. Following this, contrastive learning is employed to accentuate the distributional disparities among pixel latent vectors of different components.

As depicted in Fig. 1, the Laplace diffusion model, enhanced by contrastive learning, effectively highlights the distributional divergences between distinct components while maintaining intra-component similarity as the diffusion steps progress. The highly differentiated pixel latent vectors, which exhibit distinct evolutionary patterns, enable the segmentation model to achieve both precision and robustness with remarkable efficiency.

It is noteworthy that each image comprises $W \times H$ pixel latent vectors. The abundance and diversity of pixel latent vectors in a limited number of images suffice to capture the distributional characteristics of different components,

thereby reducing the dependency on annotated data. The pronounced differentiation in component distributions further mitigates the model's learning challenges, particularly for tail components.

Thus, our primary contribution lies in the introduction of the first Laplace diffusion model, offering a novel perspective on leveraging component distributions across diffusion steps for efficient pathology image segmentation. The devised pixel latent vector contrastive learning mechanism enhances the differentiation of component distributions, significantly improving segmentation performance, especially for tail components. A detailed theoretical analysis is provided to substantiate the practicality of the proposed L-Diffusion. Extensive experiments demonstrate that L-Diffusion achieves superior accuracy and robustness on par with existing works across multiple benchmarks.

## 2. Preliminaries

DDPM (Ho et al., 2020) consists of two processes: the forward diffusion process, which progressively adds noise to the data to transform the complex data distribution into a standard Gaussian distribution, and the counter diffusion process, which employs a parameterized $\theta$ neural network to iteratively reconstruct the data from the noise. We assume that a data $x_0$ satisfies the data distribution q($\cdot$), expressed as $x_0 \sim q(x_0)$, then the forward and backward processes can be expressed as follows:

$$q(x_t|x_{t-1}) = \mathcal{N}(x_t; \sqrt{1 - \beta_t}x_{t-1}, \beta_t\mathbf{I}), \quad (1)$$

$$p_\theta(x_{t-1}|x_t) = \mathcal{N}(x_{t-1}; \mu_\theta(x_t, t), \sigma_\theta(x_t, t)), \quad (2)$$

where $p_\theta(x_{t-1}|x_t)$ represents the new generated data distribution at $t$-th diffusion, $\beta$ represents the noise intensity, $\mu_\theta$ and $\sigma_\theta$ are the mean and variance used by the neural network to predict the distribution. Although we cannot obtain the inverted data distribution $q(x_{t-1}|x_t)$, we can obtain the conditional probability distribution $q(x_{t-1}|x_t, x_0)$ for given $x_0$ and $x_t$ according to Bayes' Theorem (3) as follows:

$$q(x_{t-1}|x_t, x_0) = q(x_t|x_{t-1}, x_0)\frac{q(x_{t-1}|x_0)}{q(x_t|x_0)}, \quad (3)$$

$$\tilde{\beta}_t = \frac{1 - \bar{\alpha}_{t-1}}{1 - \bar{\alpha}_t} \cdot \beta_t,$$

$$\tilde{\mu}_t(x_t, x_0) = \frac{\sqrt{\alpha_t}(1 - \bar{\alpha}_{t-1})}{1 - \bar{\alpha}_t}x_t + \frac{\sqrt{\bar{\alpha}_{t-1}}\beta_t}{1 - \bar{\alpha}_t}x_0, \quad (4)$$

where $\alpha_t = 1 - \beta_t$, $\bar{\alpha}_t = \prod_{i=1}^{t} \alpha_i$. From the above Eqn. (3), we extract the coefficients related to the unknown variable $x_{t-1}$ and transform them into the standard format of the probability density function of gaussian distribution. The detailed derivation can be referred to *Appendix* (A). Therefore, We can go further and get the Eqn. (4),

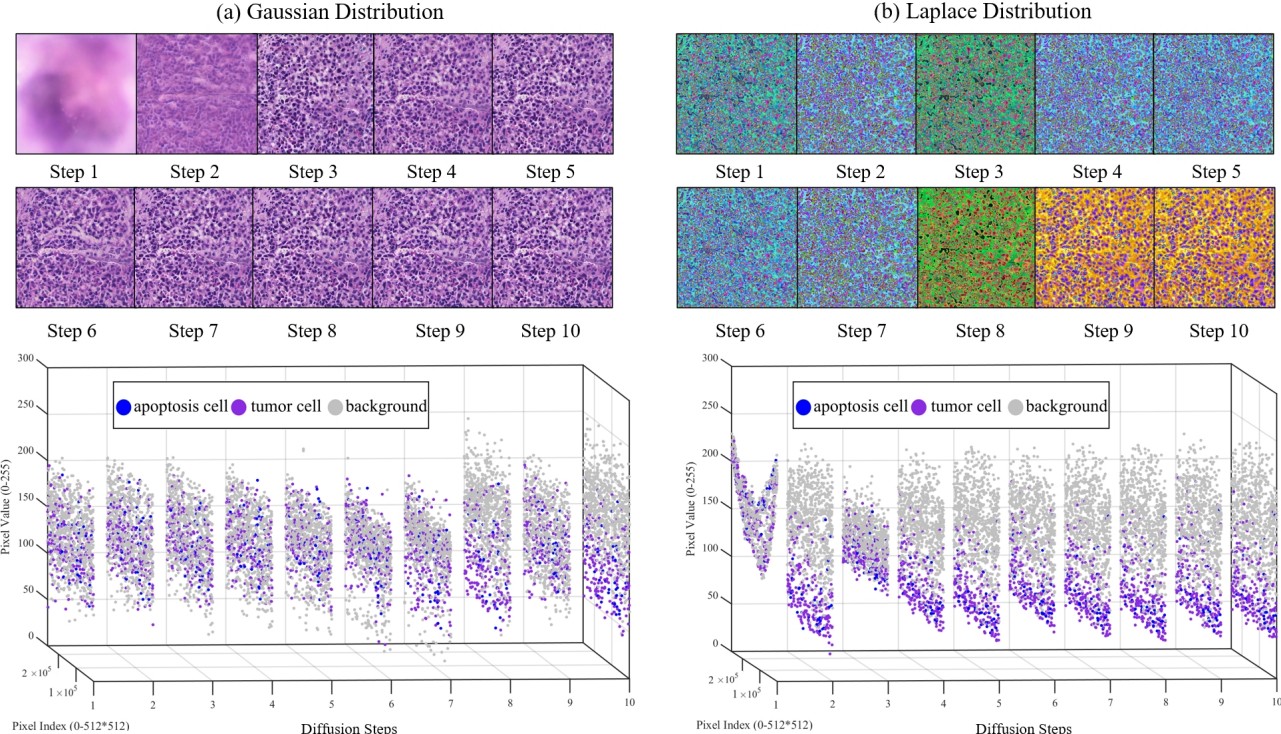

*Figure 2.* The comparison between pixel value distribution of various components modeled using Gaussian and Laplace distributions. For each diffusion step, the reconstructed image and its corresponding pixel value distribution are provided. The components modeled by Laplace distributions exhibit greater distinguishability compared to those modeled by Gaussian distributions.

Moreover, $x_t$ can be derived from $x_0$ and $\alpha_t$. Let $z_\theta(x_t, t)$ be the noise predicted by neural network at the $t$-th step noise reduction. Then we can then transform Eqn. (5) into $x_0$ denoted by $x_t$ and $z\theta(x_t, t)$ and substitute it into Eqn. (4). We can then derive the mean of the predictive distribution consisting of $x_t$ and $t$, expressed in Eqn. (6), and then calculate the generated data distribution as follows:

$$x_t = \sqrt{\bar{\alpha}_t}x_0 + \sqrt{1 - \bar{\alpha}_t} \cdot z_\theta(x_t, t), \tag{5}$$

$$\mu_\theta(x_t, t) = \frac{1}{\sqrt{\alpha_t}}(x_t - \frac{\beta_t}{\sqrt{1 - \bar{\alpha}_t}}z_\theta(x_t, t)). \tag{6}$$

## 3. Laplace Distribution Differentiation

The essence of differentiated representation of data distributions in diffusion model can be broadly categorized into two scenarios. Firstly, when two categories of data possess markedly distinct features (such as color, shape, or spatial position) the disparity in the means of their distributions should be substantial enough to minimize the overlap between the distributions. Secondly, in cases where no clear distinguishing features exist between two categories, the gradient of the distributions should be sufficiently pronounced to amplify the impact of perturbations on the probabilities, thereby enhancing the separation of the data. A more

detailed exposition of the influence of data distribution differentiation in diffusion model is provided in *Appendix* (A).

### 3.1. Laplace Distribution is More Differentiated

To differentiate the data distributions of distinct regions within the image, it is essential to select a new distribution capable of amplifying the disparities between data distributions of various categories across the entire slide, while preserving the intrinsic distribution characteristics within each component. Through extensive experimentation, we have determined that the Laplace distribution fulfills these requirements. To substantiate that this distribution is more suitable for pathology image classification than the Gaussian distribution (Gauss & Waterhouse, 2018), we will analyze it from two perspectives: theoretical derivation and qualitative comparison. Our focus lies on the gradient of the Gaussian distribution $\nabla p_\mathcal{N}(x)$ and the gradient of the Laplace distribution $\nabla p_\mathcal{L}(x)$, whose equations are as follows:

$$\nabla p_\mathcal{N}(x) = \frac{d}{dx}(\frac{1}{\sqrt{2\pi\sigma^2}}exp(-\frac{x^2}{2\sigma^2})) = -\frac{x}{\sigma^2}p_\mathcal{N}(x),$$
$$\nabla p_\mathcal{L}(x) = \frac{d}{dx}(\frac{1}{\sqrt{2b}}exp(-\frac{|x|}{b})) = -\frac{sign(x)}{b}p_\mathcal{L}(x)$$

where $b$ denotes scale parameter of Laplace distribution.

As evidenced by the preceding equation, the gradient of the Gaussian distribution exhibits proportionality to $x$, whereas the gradient of the Laplace distribution is proportional to sign$(x)$. When $x \approx 0$, $\nabla p_{\mathcal{N}}(x)$ demonstrates a smooth progression, with the gradient intensifying as $x$ diverges from the origin. In contrast, $\nabla p_{\mathcal{L}}(x)$ undergoes abrupt changes, particularly as the noise value $x$ approaches zero, rendering the Laplace distribution markedly more sensitive to its response to noise. As $x$ extends further from the origin, the Gaussian distribution exhibits a gradual decay in its tail, with $\nabla p_{\mathcal{N}}(x)$ diminishing in a relatively subdued manner. Conversely, the tail of the Laplace distribution decays more swiftly, and $\nabla p_{\mathcal{L}}(x)$ diminishes even more precipitously in the presence of substantial noise. Moreover, akin to the Gaussian distribution, which perturbs the original data by augmenting its variance, the introduction of Laplacian noise preserves the morphological structure of the original data. It merely alters the concentration of the noise distribution, signifying an enhanced sensitivity to extreme values.

Furthermore, to visually demonstrate the superiority of the Laplace distribution, we applied both Gaussian and Laplace distributions to the inverse diffusion process of the same image over 10 steps, observing the variations in pixel intensity across different regions. In the context of a melanocytoma histopathology patch, 10 sampling maps and their corresponding pixel distributions are illustrated in Fig. 2. For the pixel distribution maps, the X-axis represents the sample number (starting from 0), the Y-axis denotes the pixel count of the image, and the Z-axis indicates the range of gray values. The gray points signify the background, the purple points represent tumor cells, and the blue points denote apoptotic cells. It is evident that the pixel distribution under the Gaussian model appears more uniform. In contrast, the Laplace distribution facilitates a clearer separation of pixels belonging to distinct categories.

## 4. L-Diffusion for Efficient Pathology Image Segmentation

For the original image $x_0$, comprising $N$ distinct components, we model it using $N$ independent Laplace distributions, denoted as $q_0(x_0), q_1(x_0), \ldots, q_N(x_0)$. During the diffusion process, the mean squared error (MSE) loss for the $t$-th step is expressed as follows:

$$\mathcal{L}_{MSE} = \mathbb{E}_{x_0, \epsilon, t} \left[ \| \epsilon - \epsilon_\theta(x_t, t) \|^2 \right],$$

where $\epsilon$ represents the added Laplace noise, and $\epsilon_\theta(x_t, t)$ denotes the predicted noise derived from the intermediate reconstructed feature map $x_t$ using model parameters $\theta$. For clarity in presenting the methodology, we omit the detailed

derivation of the multiple-component Laplace distribution modeling, which can be found in *Appendix* (A).

Through $T$ diffusion steps, we obtain a sequence of intermediate reconstructed feature maps $[r_1, r_2, \ldots, r_t, \ldots, r_T]$, where $r_t \in \mathbb{R}^{W \times H \times 3}$, and $W$ and $H$ represent the width and height of $x_0$, respectively. Each reconstructed feature map $r_t$ is subsequently transformed into a grayscale image $\hat{r}_t = \text{RGB2Gray}(r_t)$, forming a sequence of feature maps $[\hat{r}_1, \hat{r}_2, \ldots, \hat{r}_t, \ldots, \hat{r}_T]$, with $\hat{r}_t \in \mathbb{R}^{W \times H \times 1}$. Along the diffusion step axis, each pixel corresponds to a vector $v \in \mathbb{R}^{1 \times T}$, referred to as the pixel latent vector.

**Pixel Latent Vector Contrastive Learning**. To enhance the distributional similarity among positive pairs Positive$(v_k^n, v_{k'}^n)$ and the distributional disparities among negative pairs Negative$(v_k^n, v_i^{n'})$, contrastive learning is employed. The contrastive learning loss function $\mathcal{L}_{CRT}^{k,k'}$ is defined as follows:

$$\mathcal{L}_{CRT}^{k,k'} = -\log \frac{\text{Positive}(v_k^n, v_{k'}^n)}{\text{Positive}(v_k^n, v_{k'}^n) + \text{Negative}(v_k^n, v_i^{n'})},$$

$$\text{Negative}(v_k^n, v_i^{n'}) = \sum_{n'=1}^{N} \sum_{i=1}^{K_{n'}} \exp(\text{sim}(v_k^n, v_i^{n'})/\tau),$$

$$\text{Positive}(v_k^n, v_{k'}^n) = \exp(\text{sim}(v_k^n, v_{k'}^n)/\tau), \quad n \neq n',$$

where $N$ represents the number of components, and $K_{n'}$ denotes the sampling number of vector samples for the $n'$-th component.

By combining the MSE loss $\mathcal{L}_{MSE}$ and the contrastive learning loss $\mathcal{L}_{CRT}^{k,k'}$, the distributions of various components in the intermediate feature maps become highly differentiated, as illustrated in Fig. 1. This differentiation significantly reduces the identification difficulty for the subsequent segmentation module. It is important to note that the contrastive learning loss $\mathcal{L}_{CRT}^{k,k'}$ is applicable only to a subset of pathology images with available annotations.

**Sequence Feature Enhanced Segmentation**. Utilizing the enhanced feature maps $[\hat{r}_1', \hat{r}_2', \ldots, \hat{r}_t', \ldots, \hat{r}_T']$ refined through contrastive learning, the segmentation network $\mathcal{F}_{seg}()$ generates the prediction results $y'$ as follows:

$$y' = \mathcal{F}_{seg}([\hat{r}_1', \hat{r}_2', \ldots, \hat{r}_t', \ldots, \hat{r}_T']).$$

The DICE loss between the predicted map $y'$ and the ground truth $y$ is employed to train the segmentation network, defined as:

$$\mathcal{L}_{\text{Dice}}(y', y) = 1 - \frac{2y' \cdot y}{y' + y}.$$

Through the two-stage training framework combining L-Diffusion and the segmentation network, the pathology image is effectively segmented into its distinct components.

*Table 1.* Quantitative tissue segmentation performance (%) comparison. **Bold** and underling indicate the best and second-best performance.

| MODEL | CRCD | | | | PUMA | | | | BCSS | | | |
|---|---|---|---|---|---|---|---|---|---|---|---|---|
| | DICE | MPA | MIoU | FwIoU | DICE | MPA | MIoU | FwIoU | DICE | MPA | MIoU | FwIoU |
| FASTFCN | 45.23 | 44.68 | 46.11 | 76.20 | 48.77 | 45.09 | 50.72 | 67.60 | 45.41 | 51.53 | 46.78 | 71.96 |
| U-NET++ | 73.02 | 51.12 | 66.18 | 79.11 | 78.07 | 53.46 | 66.51 | 82.26 | 76.04 | 53.31 | 61.23 | 74.91 |
| SWIN-UNET | 70.08 | 52.22 | 63.27 | 84.11 | 69.46 | 52.39 | 55.40 | 80.42 | 66.24 | 52.85 | 61.54 | 84.32 |
| SAMUS | 63.94 | 62.45 | 55.82 | 77.93 | 61.24 | 61.29 | 53.92 | 82.91 | 62.68 | 59.26 | 53.34 | 79.58 |
| SAMED | 70.63 | 50.38 | 58.38 | 78.08 | 62.56 | 57.39 | 64.48 | 70.74 | 67.51 | 51.71 | 56.48 | 75.79 |
| SAMPATH | 77.54 | 66.70 | 63.65 | 86.28 | 84.95 | 57.79 | 66.30 | 83.92 | 78.10 | 62.83 | 67.44 | 87.12 |
| UN-SAM | 75.69 | 66.30 | 62.71 | 83.57 | 80.49 | 56.63 | 65.19 | 85.73 | 80.89 | 65.80 | 70.44 | 85.38 |
| DEEPLABV3 | 73.12 | 58.49 | 49.18 | 82.26 | 73.36 | 53.00 | 53.51 | 81.43 | 67.75 | 52.76 | 57.83 | 87.62 |
| DEEPLABV3+ | 77.04 | 58.14 | 52.80 | 81.88 | 80.85 | 58.41 | 48.01 | 88.55 | 86.31 | 56.53 | 50.19 | 84.30 |
| DENSEASPP | 76.18 | 67.39 | 60.50 | 77.71 | 76.24 | 60.02 | 59.33 | 72.77 | 70.88 | 64.53 | 54.68 | 81.35 |
| BONUS | 75.22 | 71.69 | 64.59 | 80.30 | 72.13 | 60.64 | 68.77 | 83.14 | 75.07 | 70.19 | 60.31 | 86.73 |
| L-DIFFUSION | **82.38** | **80.19** | **80.17** | **86.47** | **92.11** | **88.03** | **81.62** | **91.77** | **89.24** | **82.33** | **83.49** | **91.17** |
| *Improvement* | *+4.84* | *+12.80* | *+16.52* | *+0.19* | *+7.16* | *+26.74* | *+15.11* | *+3.22* | *+2.93* | *+17.80* | *+16.05* | *+3.55* |

*Table 2.* Quantitative evaluation of cellular segmentation performance (%). Metric calculations are only performed on foreground cells.

| MODEL | CRCD | | | | PUMA | | | | PANNUKE | | | |
|---|---|---|---|---|---|---|---|---|---|---|---|---|
| | DICE | MPA | MIoU | FwIoU | DICE | MPA | MIoU | FwIoU | DICE | MPA | MIoU | FwIoU |
| FASTFCN | 43.04 | 60.25 | 32.86 | 73.41 | 39.03 | 62.87 | 31.28 | 74.15 | 44.51 | 68.73 | 30.58 | 66.52 |
| U-NET++ | 73.73 | 82.17 | 30.06 | 65.93 | 74.02 | 75.34 | 34.68 | 75.17 | 36.88 | 81.96 | 25.89 | 71.09 |
| SWIN-UNET | 46.70 | 27.70 | 31.54 | 86.12 | 40.05 | 29.89 | 30.94 | 80.47 | 47.08 | 32.94 | 75.76 | 79.71 |
| SAMUS | 34.53 | 75.50 | 37.72 | 75.34 | 32.26 | 77.10 | 37.52 | 72.04 | 30.11 | 70.17 | 37.27 | 68.66 |
| SAMED | 70.85 | 61.21 | 34.93 | 69.66 | 66.88 | 64.57 | 31.93 | 73.02 | 72.60 | 61.33 | 26.59 | 67.95 |
| SAMPATH | 79.02 | 87.97 | 82.03 | 81.23 | 81.62 | 80.79 | 74.50 | 80.68 | 72.33 | 82.79 | 38.08 | 80.46 |
| UN-SAM | 78.53 | 85.27 | 81.79 | 80.36 | 79.57 | 74.99 | 68.81 | 79.12 | 70.48 | 80.77 | 75.42 | 84.57 |
| DEEPLABV3 | 54.65 | 52.12 | 52.54 | 84.93 | 56.19 | 55.55 | 45.07 | 81.83 | 53.98 | 60.82 | 52.45 | 82.93 |
| DEEPLABV3+ | 63.43 | 59.01 | 71.66 | 79.38 | 65.99 | 55.92 | 75.00 | 82.59 | 71.46 | 55.17 | 72.41 | 87.30 |
| DENSEASPP | 37.35 | 48.09 | 51.09 | 84.41 | 38.90 | 52.33 | 51.47 | 81.66 | 81.00 | 54.95 | 45.46 | 77.54 |
| BONUS | 75.33 | 68.67 | 60.42 | 79.84 | 78.44 | 75.49 | 64.53 | 82.14 | 80.33 | 60.58 | 69.22 | 85.97 |
| L-DIFFUSION | **96.11** | **94.94** | **92.62** | **95.34** | **88.57** | **90.38** | **86.19** | **83.13** | **94.78** | **93.46** | **90.18** | **92.18** |
| *Improvement* | *+20.09* | *+6.97* | *+10.59* | *+10.41* | *+6.95* | *+9.59* | *+11.19* | *+0.54* | *+13.78* | *+10.67* | *+14.42* | *+4.88* |

# 5. Experiment

**Datasets and Evaluation Metrics**. We employ six distinct tissue and cellular datasets to validate the multi-scale segmentation capabilities of L-Diffusion. These datasets encompass: a colorectal cancer histopathology dataset provided by the Guangdong Provincial People's Hospital (referred to as CRCD) (Ye et al., 2023), a melanoma histopathology dataset from the PUMA challenge (referred to as PUMA) (Schuiveling et al., 2024), a publicly accessible dataset specifically curated for tissue segmentation tasks in breast cancer pathology (referred to as BCSS) (Amgad et al., 2019), and a publicly available dataset dedicated to multi-class cellular segmentation (referred to as PanNuke) (Gamper et al., 2020). Furthermore, comprehen-

sive details regarding these datasets are provided in Table 7 of *Appendix (B)*. The evaluation metrics comprise MPA, DICE, mIoU, and FWIoU, which collectively evaluate segmentation performance from the perspectives of pixel-wise accuracy, segmentation area, boundary precision, and class-specific analysis. It's noticed that the performance on cell only considering the cells, the reason of which is that background has large proportion of whole slide image will disturb the cells segmentation performance measurement.

**Models and Parameters**. L-Diffusion comprises a VAE encoder, a Laplace Scheduler, a U-Net noise prediction module, a DAE decoder, and a contrastive learning module. The image is initially encoded into the latent space by the VAE, after which it is passed to the Laplace Scheduler along with

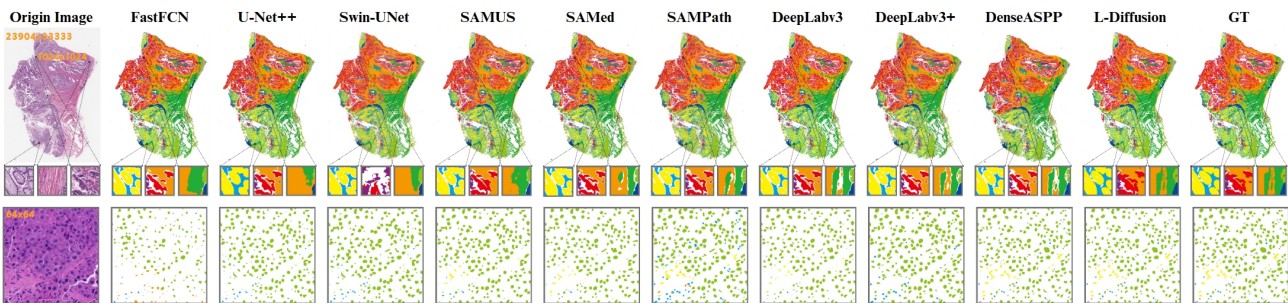

*Figure 3.* Qualitative comparison of tissue (upper row) and cellular (lower row) segmentation performance.

*Table 3.* The ablation study on annotation ratio.

| Annotation Ratio | 10% | 20% | 30% | 50% | 70% | 100% |
|---|---|---|---|---|---|---|
| DICE | 19.95 | 59.59 | 83.51 | 86.19 | 91.62 | 92.11 |
| MPA | 20.00 | 55.12 | 82.93 | 85.73 | 87.95 | 88.03 |
| mIoU | 19.91 | 54.96 | 79.86 | 80.02 | 80.54 | 81.62 |
| FwIoU | 33.06 | 71.54 | 89.46 | 89.67 | 90.31 | 91.77 |

*Table 4.* Qualitative ablation study results.

| Contrastive Learning? | Data Distribution | Average Score on Tissue | | | |
|---|---|---|---|---|---|
| | | DICE | MPA | mIoU | FwIoU |
| ✘ | Gaussian | 15.33 | 14.17 | 13.31 | 23.26 |
| ✓ | Gaussian | 17.41 | 21.61 | 16.91 | 29.76 |
| ✘ | Laplace | 26.02 | 26.81 | 21.41 | 34.26 |
| ✓ | Laplace | 85.75 | 83.52 | 81.76 | 89.74 |

the time step $t$ to generate noise. This noisy representation is then processed by the U-Net to predict the noise at step $t$, which is subsequently subtracted from the noisy image to yield the latent space sampling. The resulting latent representation is fed into the DAE to decode it back to its original dimensions. $K_{n'}$ is set to 100. We adopt the ConvNeXT (Z. Liu et al., 2022) as the segmentation network. To Train Diffusion Model, we configure the batch size to 1, employ the Adam optimizer with a learning rate of $1 \times 10^{-5}$, and typically set the number of sampling steps to $5 \sim 15$, contingent upon the available GPU. In the contrastive learning module, the temperature $\tau$ ranges from $0.05 \sim 0.1$ to ensure effective sharpening of the distribution without gradient explosion. In addition, to train ConvNeXT, we configure the batch size to 32, employ the Adam optimizer with a learning rate of $1 \times 10^{-3}$.

## 5.1. Comparison with SOTA Methods

In order to comprehensively verify the performance of the proposed model, we compared it with a variety of existing mainstream models including FastFCN (H. Wu et al., 2019), U-Net++ (Zhou et al., 2018), Swin-UNet (Cao et al., 2022), SAMUS (X. Lin et al., 2023), SAMed (K. Zhang & Liu, 2023), SAMPath (J. Zhang et al., 2023), DeepLabv3 (L.-C. Chen, 2017), DeepLabv3+ (L.-C. Chen et al., 2018), DenseASPP (Yang et al., 2018), UN-SAM (Z. Chen et al., 2024) and BONUS (Y. Lin et al., 2024).

**Quantitative Comparison**. The quantitative results for tissue and cellular segmentation are detailed in Table 1&2. L-Diffusion demonstrates a significant enhancement on both tissue and cell segmentation datasets. These substantial

gains effectively validate the efficacy of leveraging component latent distributions for pathology image segmentation.

**Qualitative Comparison**. Fig. 3 presents the qualitative visualization, illustrating that L-Diffusion attains exceptional boundary segmentation accuracy in the tissue samples depicted in the upper row. For cellular samples, L-Diffusion not only achieves precise boundary segmentation but also excels in accurately identifying tail-class cells, as highlighted by the yellow and cyan annotations. Additional visualization results are provided in *Appendix* (D)-(F).

**Performance for Distinct Components**. The radar chart illustrates the quantitative assessment of segmentation performance across various methodologies for distinct tissue and cellular categories. As depicted in Fig. 4, it is evident that L-Diffusion outperforms all existing methods across every tissue and cellular category, particularly excelling in the segmentation of components with lower proportions. This underscores L-Diffusion's proficiency in addressing long-tail distribution challenges within pathology images.

## 5.2. Performance Across Varied Annotation Ratios

To validate the efficiency of L-Diffusion under limited annotations, we conduct an ablation study using varying annotation ratios on the PUMA tissue segmentation dataset. As illustrated in Table 3, it is evident that L-Diffusion achieves comparable performance even with only 30% of annotated samples. However, the segmentation module necessitates a sufficient number of annotated samples, thereby increasing the annotation demands of L-Diffusion. In future work, we aim to explore the minimal annotation data requirements.

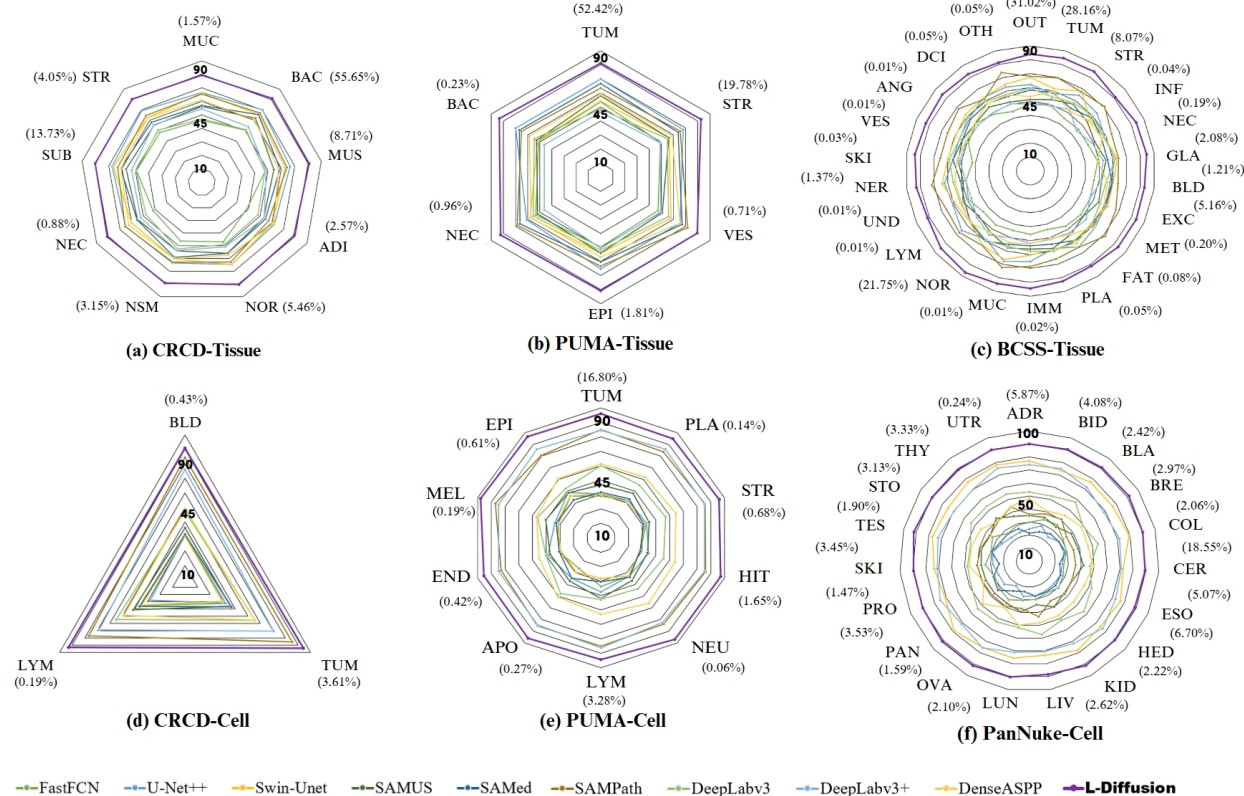

*Figure 4.* A quantitative evaluation of segmentation performance across diverse methodologies for distinct tissue and cellular categories. Each axis of the radar chart represents a specific component category along with its corresponding proportion.

## 5.3. Ablation Study

In this section, we conduct an extensive series of ablation studies on the critical components of L-Diffusion, including the Laplace distribution and pixel latent vector contrastive learning. Table 4 presents the average performance metrics across three tissue datasets, revealing that the integration of the Laplace distribution with contrastive learning serves as the cornerstone of L-Diffusion for pathology image segmentation. Fig. 5 illustrates that the latent feature distribution under the "contrastive learning + Laplace distribution" configuration exhibits more pronounced distinctions between different tissue types. As the diffusion steps progress, the data distribution within the same tissue type becomes increasingly cohesive, further validating the efficacy of combining contrastive learning with the Laplace distribution.

## 5.4. Generalization on Other Large-scale Image

To evaluate the generalization prowess of L-Diffusion across other large-scale image datasets, we compare its segmentation performance with several state-of-the-art (SOTA) remote sensing image segmentation methods on the

Massachusetts-Building dataset (Mnih, 2013). The competing methods encompass U-Net (Ronneberger et al., 2015), Uniformer (K. Li et al., 2023), UANet (J. Li et al., 2024), and GLGF-Net (Fu et al., 2024). For a fair comparison, we employ widely recognized metrics in the remote sensing image segmentation domain, including IoU, F1, and Precision (Pre.). As illustrated in Table 6, L-Diffusion exhibits superior performance in remote sensing image segmentation, rivaling existing methodologies. Segmentation visualization results, detailed in *Appendix* (G), further demonstrate that L-Diffusion achieves comparable segmentation accuracy, particularly for tail-class objects such as buildings. These experiments highlight the robust generalization and segmentation capabilities of L-Diffusion on large-scale images, especially for tail-class categories.

## 5.5. Comparison with Different Distributions

We investigate the influence of different noise distributions on the average performance of diffusion models on three tissue segmentation datasets, focusing on three representative types: Cauchy (Lian et al., 2025), Student's t (X.-F. Wang et al., 2024), and Laplace. As shown in Table 5, the Laplace distribution achieves the best overall performance. In con-

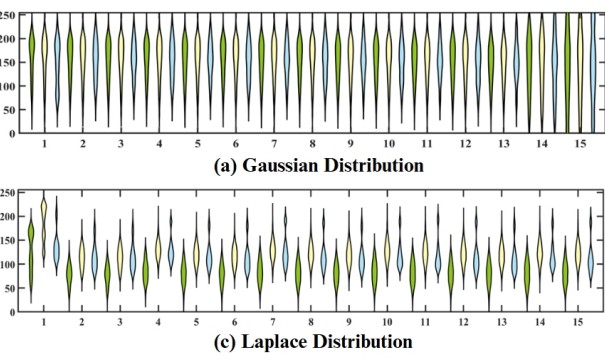
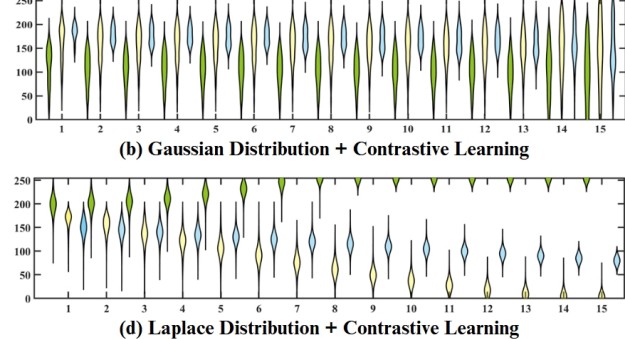

*Figure 5.* A comparative analysis of latent feature distributions under various configurations. The latent feature distributions for individual components across multiple diffusion steps are represented by violin plots, distinguished by green, yellow, and cyan colors.

trast, the Cauchy distribution yields a significantly lower DICE score and FwIoU score. This can be attributed to its undefined expectation and variance, which severely hinders the stability of gradient-based optimization. Student's t distribution demonstrates promising segmentation accuracy, but suffers from a prohibitively high computational cost, likely due to its more complex gradient behavior. Overall, the Laplace distribution maintains a strong balance between segmentation accuracy and computational efficiency.

*Table 5.* Comparison of different noise distributions on segmentation performance.

| Distribution | DICE (%) | FwIoU (%) | Runtime (s) |
|---|---|---|---|
| Cauchy | 16.25 | 20.33 | 7325 |
| Student's $t$ | 83.17 | 86.50 | 20440 |
| Laplace | **85.75** | **89.80** | **8882** |

*Table 6.* Generalization on remote sensing image dataset.

| Metric | U-Net | Uniformer | UANet | GLGF-Net | L-Diffusion |
|---|---|---|---|---|---|
| IoU | 68.48 | 73.80 | 76.41 | 75.33 | 80.30 |
| F1 | 81.47 | 84.92 | 86.63 | 85.93 | 88.98 |
| Pre. | 80.09 | 87.60 | 87.94 | 85.03 | 96.52 |

## 6. Related Work

**Pathology Image Segmentation** plays a pivotal role in the diagnosis and treatment of cancer, while also fostering a deeper comprehension of the mechanisms underlying tumorigenesis, progression, and metastasis. Numerous studies have utilized various segmentation models, such as U-Net (Ronneberger et al., 2015), DeepLab (L.-C. Chen et al., 2017) and Transformer (Atabansi et al., 2023), along with their variants, for pathology image segmentation (Y. Wu et al., 2022). Based on their segmentation objectives, these studies can be broadly classified into two categories: cell segmentation (Ciresan et al., 2012; D. Liu et al., 2019;

Hayakawa et al., 2021; Feng et al., 2021; Mahbod et al., 2024) and tissue segmentation (Salvi et al., 2021; Musulin et al., 2021; Ye et al., 2023). However, gigapixel pathology images exhibit multi-scale features and long-tail components, which result in significant annotation labor costs and limited accuracy in identifying tail-class samples.

**Contrastive Learning**, a pivotal approach in self-supervised learning, has achieved remarkable advancements in the field of computer vision in recent years. Researchers have extended the application of contrastive learning to diverse domains, including natural language processing (NLP), graph learning, and multi-modal learning (Le-Khac et al., 2020; Hu et al., 2024). Recently, several studies have integrated contrastive learning into diffusion models (Tian et al., 2024; Dalva & Yanardag, 2024; Xiao et al., 2024), utilizing it to amplify the representational distance between different classes. In this paper, we apply contrastive learning to enhance the distinction between pixel latent vectors of different components modeled using Laplace distributions.

**Diffusion in Medical Image Analysis.** Wu et al. (J. Wu, Fu, et al., 2024; J. Wu, Ji, et al., 2024) introduced MedSegDiff, a diffusion model-based approach for medical image segmentation. Webber et al. (Webber & Reader, 2024) utilized diffusion models for medical image reconstruction, enabling the production of high-quality images even in low-dose or rapid imaging scenarios. Kazerouni et al. (Kazerouni et al., 2023) given a comprehensive survey of diffusion models in medical imaging. Additionally, several researchers (Oh & Jeong, 2023; He et al., 2024; Xu et al., 2024) have extended the application of diffusion models to pathology image generation tasks, leveraging their exceptional capability for high-quality image synthesis.

**Diffusion in Segmentation.** Amit et al. (Amit et al., 2021) pioneered the integration of diffusion models into segmentation tasks, proposing the SegDiff model and exploring the application of diffusion models to address image segmentation challenges. Bogensperger et al. (Bogensperger

et al., 2023) introduced an innovative image segmentation approach by merging score-based models with diffusion models. Wang et al. (M. Wang et al., 2023) developed SegRefiner, which employs a discrete diffusion process to iteratively refine segmentation results. Xie et al. (Xie et al., 2024) utilized diffusion models to generate synthetic labeled data for segmentation tasks. To the best of our knowledge, no existing method has leveraged component distributions across diffusion steps for segmentation as we have accomplished.

**Distribution Exploration** Recent studies have explored how different noise distributions affect diffusion model performance. Hang et al. (Hang et al., 2024) show that the Laplace distribution, with its sharp peak and heavy tails, improves robustness to outliers. Alexia et al. (Jolicoeur-Martineau et al., 2023) further compare Gaussian, T, and Uniform distributions, finding that smoother distributions lead to better image generation. However, these works mainly focus on generation quality. In contrast, this paper is the first to explore how sharp distributions can reshape the latent space for segmentation tasks. We provide a detailed theoretical analysis and show that distribution-aware design benefits segmentation performance in diffusion models.

## 7. Conclusion

In this paper, we introduce L-Diffusion, an innovative framework that employs Laplace distributions and contrastive learning to advance pathology image segmentation. By modeling distinct components using Laplace distributions, L-Diffusion amplifies distributional divergences, facilitating precise and robust segmentation. The proposed pixel latent vector contrastive learning mechanism diminishes the dependency on annotated data and alleviates the learning challenges associated with tail components. Rigorous theoretical analysis and comprehensive experimental evaluations substantiate the practicality and superiority of L-Diffusion, positioning it as a state-of-the-art solution for pathology image segmentation. The Laplace Diffusion Model offers a groundbreaking perspective on harnessing component distributions across diffusion steps to achieve efficient pathology image segmentation. It provides an effective tool to aid tumor diagnosis and microenvironment analysis, fostering a deeper understanding of tumorigenesis and progression. In future work, we aim to explore the application of Laplace Diffusion in an unsupervised manner to assist pathology image component analysis and extend its capabilities to other large-scale image segmentation tasks.

## Ethical and Clinical Statements

This study explores the use of L-Diffusion for academic research in tumor diagnosis and microenvironment analy-

sis. While our method demonstrates strong technical potential, we acknowledge broader ethical considerations beyond dataset access. These include the possibility of algorithmic bias affecting diagnostic outcomes across different patient groups, challenges in model interpretability for clinical practitioners, and concerns related to patient data privacy during inference and deployment.

The current version of L-Diffusion is intended solely for academic investigation and is not deployed in clinical settings. Any future application in real-world scenarios should be guided by clinical validation, domain-specific oversight, and compliance with relevant medical and regulatory standards. We advocate for continued research into the safe, fair, and transparent use of AI systems in healthcare.

## Impact Statement

Regarding the dataset, all utilized pathology datasets are publicly available and have undergone ethical review and approval. Concerning the algorithm, L-Diffusion delivers precise and robust pathology image segmentation with exceptional efficiency. Its capacity to effectively decompose pathology images simplifies the analytical workflow, providing a potent tool for advancing tumor diagnosis and microenvironment analysis. This progress cultivates a more profound comprehension of the mechanisms driving tumorigenesis and progression.

## Acknowledgements

This work is supported by National Natural Science Foundation of China (62376248), the Huadong Medicine Joint Fund of the Zhejiang Provincial Natural Science Foundation of China (LHDMZ25H160002), the Zhejiang Province Health Major Science and Technology Program of National Health Commission Scientific Research Fund (No. WKJ-ZJ-2426) and Information Technology Center, ZheJiang University.

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

## A. Mathematical Derivations

This section provides the important mathematical derivations mentioned in *Section 4*. As we assume, the data distribution satisfied by the original image $x_0$ can be expressed as the following formula (7):

$$q(x_0|\mu, b) = \sum_{n=1}^{N} q_n(x_0|\mu_n, b_n), q(x_{1:T}|x_0) = \prod_{t=1}^{T} q(x_t|x_{t-1}) \tag{7}$$

In the above formula, $N$, $\mu_i$ and $b_i$ are the number of categories and the data distribution center and scale parameters of each category, and $t$ and $T$ represent the current time step and the total step number respectively.

In order to propagate the gradient backwards, we need to make the $z$ sampled by Laplace noise differentiable through a heavy parameter operation. It is possible to set an independent variable $\epsilon$, when the sampling noise $z$ satisfies $z = \mu_\theta + b_\theta \odot \epsilon$, it is observed that the equation still maintains randomness as a whole, and satisfies the Laplace distribution with the mean $\mu_\theta$ and scale parameter $b_\theta$. The subsequent inference stages $\mu_\theta$ and $b_\theta$ can be derived from U-Net with parameter $\theta$, and the randomness is transferred to $\epsilon$, making the gradient derivable.

$$
\begin{aligned}
x_t &= \sum_{n=1}^{N} (\sqrt{\alpha_{(n,t)}} x_{t-1} + \sqrt{1 - \alpha_{(n,t)}} z_1), \\
&= \sum_{n=1}^{N} \left( \sqrt{\alpha_{(n,t)}}(\sqrt{\alpha_{(n,t-1)}} x_{t-2} + \sqrt{1 - \alpha_{(n,t-1)}} z_2) + \sqrt{1 - \alpha_{(n,t)}} z_1 \right), \\
&= \sum_{n=1}^{N} \left( \sqrt{\alpha_{(n,t)}\alpha_{(n,t-1)}} x_{t-2} + (\sqrt{\alpha_{(n,t)}(1 - \alpha_{(n,t-1)})} z_2 + \sqrt{1 - \alpha_{(n,t)}} z_1) \right).
\end{aligned}
\tag{8}
$$

The above formula is $x_{t-1}$ plus noise $z_1$ to obtain a mathematical representation of $x_t$, where $\alpha_{(n,t)}$ represents the noise level coefficient of class $n$ at $t$ time steps, equivalent to $1 - \beta(n,t)$ and satisfies $\bar{\alpha}_{(n,t)} = \prod_{t=1}^{T} \alpha_{(n,t)}$. Since Laplace distribution and Gaussian distribution have the same additivity, the formula (9) can be derived as follows:

$$
\begin{aligned}
\sqrt{\alpha_{(n,t)}(1 - \alpha_{(n,t-1)})} z_2 + \sqrt{1 - \alpha_{(n,t)}} z_1 &= \sqrt{\alpha_{(n,t)}(1 - \alpha_{(n,t-1)}) + 1 - \alpha_{(n,t)}} \bar{z}_2, \\
&= \sqrt{1 - \alpha_{(n,t)}\alpha_{(n,t-1)}} \bar{z}_2.
\end{aligned}
\tag{9}
$$

Substituting formula (9) into formula (8) gives the formula (10) as follows:

$$
\begin{aligned}
x_t &= \sum_{n=1}^{N} \left( \sqrt{\alpha_{(n,t)}\alpha_{(n,t-1)}} x_{t-2} + \sqrt{1 - \alpha_{(n,t)}\alpha_{(n,t-1)}} \bar{z}_2 \right), \\
&= \sum_{n=1}^{N} \left( \sqrt{\alpha_{(n,t)}\alpha_{(n,t-1)}}(\sqrt{\alpha_{(n,t-2)}} x_{t-3} + \sqrt{1 - \alpha_{(n,t-2)}} z_3) + \sqrt{1 - \alpha_{(n,t)}\alpha_{(n,t-1)}} \bar{z}_2 \right), \\
&= ..., \\
&= \sum_{n=1}^{N} \left( \sqrt{\prod_{j=1}^{t} \alpha_{(n,j)}} x_0 + \sqrt{1 - \prod_{j=1}^{t} \alpha_{(n,j)}} \bar{z}_t \right), \\
&= \sum_{n=1}^{N} (\sqrt{\bar{\alpha}_{(n,t)}} x_0 + \sqrt{1 - \bar{\alpha}_{(n,t)}} \bar{z}_t).
\end{aligned}
\tag{10}
$$

Equation (11) can be obtained by substituting the above equation into Laplace distribution, and the noise adding process can be obtained after transformation as follows:

$$
\begin{aligned}
q(x_t|x_{t-1}) &= \sum_{n=1}^{N} L_n(x_t; \sqrt{\bar{\alpha}_{(n,t)}} x_{t-1}, 1 - \bar{\alpha}_{(n,t)} \mathbf{I}), \\
&= \sum_{n=1}^{N} L_n(x_t; \sqrt{1 - \bar{\beta}_{(n,t)}} x_{t-1}, \bar{\beta}_{(n,t)} \mathbf{I}).
\end{aligned}
\tag{11}
$$

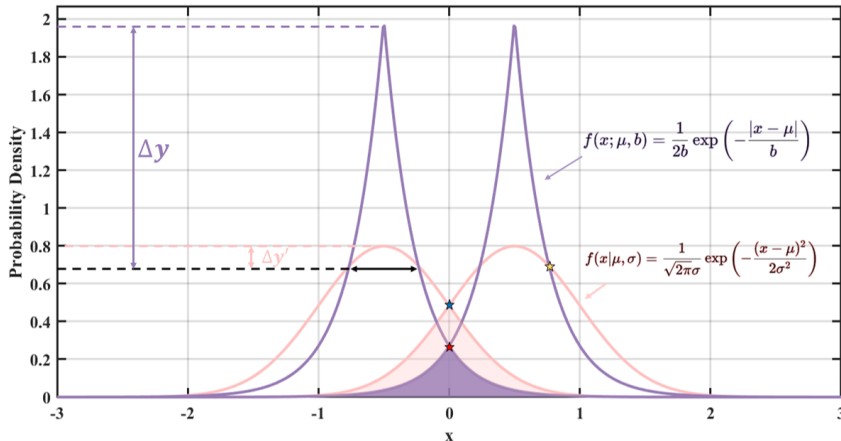

*Figure 6.* Illustration of Laplace and Gaussian distributions. The purple curve represents the Laplace distribution with means of $-1$ and $1$, respectively, and a scale parameter of $0.25$, while the pink curve depicts the Gaussian distribution with means of $-1$ and $1$ and a variance of $0.5$.

In the process of reverse diffusion, we still refer to formulas (3). Therefore, backward diffusion can be expressed as the formula (12):

$$p_\theta(x_{0:T}) = p(x_T) \prod_{t=1}^{T} p_\theta(x_{t-1}|x_t), \, p_\theta(x_{t-1}|x_t) = \sum_{i=1}^{N} L_i(x_{t-1}; \mu_\theta(x_t, t), b_\theta(x_t, t)). \quad (12)$$

The formula for conditional probability $P(A|B)$ with marginal probabilities $P(A)$ and $P(B)$ can be expressed as $P(A|B) = \dfrac{P(A \cap B)}{P(B)}$. And then $P(A \cap B) = P(A|B) \cdot P(B) = P(B|A) \cdot P(A)$. Therefore, we take the "perturbation" sampled data $\tilde{x}_{t-1}$ as A, the complete Laplace noise and the original data $x_t$ as B to obtain the formula (13):

$$q(\tilde{x}_{t-1}|x_t) = \frac{q(x_t|\tilde{x}_{t-1}) \cdot q(\tilde{x}_{t-1})}{q(x_t)}$$

$$q(\tilde{x}_{t-1}|x_t, x_0) = \frac{q(x_t|\tilde{x}_{t-1}, x_0) \cdot q(\tilde{x}_{t-1}|x_0)}{q(x_t|x_0)} \quad (13)$$

Because the conditional probability in formula (13) satisfies formula (12), formula (14) can be derived as follows:

$$q(\tilde{x}_{t-1}|x_t, x_0) = \frac{q(x_t|\tilde{x}_{t-1}, x_0) \cdot q(\tilde{x}_{t-1}|x_0)}{q(x_t|x_0)},$$

$$\Rightarrow -\sum_{n=1}^{N} \left( \frac{|x_t - \sqrt{\alpha_{(n,t)}}\tilde{x}_{t-1}|}{\sqrt{\beta_{(n,t)}}} + \frac{|\tilde{x}_{t-1} - \sqrt{\bar{\alpha}_{(n,t-1)}}x_0|}{\sqrt{1 - \bar{\alpha}_{(n,t-1)}}} - \frac{|x_t - \sqrt{\bar{\alpha}_{(n,t)}}x_0|}{\sqrt{1 - \bar{\alpha}_{(n,t)}}} \right). \quad (14)$$

Before, we derived a formula (8). As a result, know $x_t = \sum_{n=1}^{N}(\sqrt{\alpha_{(n,t)}}x_{t-1} + \sqrt{1 - \alpha_{(n,t)}}z_1)$. And because $\sqrt{1 - \alpha_{(n,t)}}z_1$ is always non-negative, so it is not difficult to draw $x_t - \sqrt{\alpha_{(n,t)}}\tilde{x}_{t-1} \geq 0$. Similarly, we can also draw $\tilde{x}_{t-1} - \sqrt{\bar{\alpha}_{(n,t-1)}}x_0 \geq 0$ and $x_t - \sqrt{\bar{\alpha}_{(n,t)}}x_0 \geq 0$. Substitute the formula (14) to get formula (15) as follows:

$$q(\tilde{x}_{t-1}|x_t, x_0) \Rightarrow -\sum_{n=1}^{N} \left( \frac{|x_t - \sqrt{\alpha_{(n,t)}}\tilde{x}_{t-1}|}{\sqrt{\beta_{(n,t)}}} + \frac{|\tilde{x}_{t-1} - \sqrt{\bar{\alpha}_{(n,t-1)}}x_0|}{\sqrt{1 - \bar{\alpha}_{(n,t-1)}}} - \frac{|x_t - \sqrt{\bar{\alpha}_{(n,t)}}x_0|}{\sqrt{1 - \bar{\alpha}_{(n,t)}}} \right),$$

$$= -\sum_{n=1}^{N} \left( \frac{x_t - \sqrt{\alpha_{(n,t)}}\tilde{x}_{t-1}}{\sqrt{\beta_{(n,t)}}} + \frac{\tilde{x}_{t-1} - \sqrt{\bar{\alpha}_{(n,t-1)}}x_0}{\sqrt{1 - \bar{\alpha}_{(n,t-1)}}} - \frac{x_t - \sqrt{\bar{\alpha}_{(n,t)}}x_0}{\sqrt{1 - \bar{\alpha}_{(n,t)}}} \right),$$

$$= -\sum_{n=1}^{N} \left( \left( -\sqrt{\frac{\alpha_{(n,t)}}{\beta_{(n,t)}}} + \frac{1}{\sqrt{1 - \alpha_{n,t-1}}} \right) \tilde{x}_{t-1} + \boxed{C(x_t, x_0)} \right). \quad (15)$$

The red box refers to the scale parameter of the Laplace distribution, while the blue box refers to the mean value of the Laplace distribution. The data distribution for category $n$ can be obtained after transformation.

In addition, since the above random variable $\epsilon_{(i,\theta)}(x_{(i,t)})$ satisfies the Laplace distribution, it can be found by comparing Fig. 6. Laplace probability change range $\Delta y$ much larger than Gaussian distribution probability change range $\Delta y'$. Observing the red pentagrams representing intersection points of different means in the Laplace distribution and the blue pentagrams for the Gaussian distribution, it is evident that the red pentagrams occur with lower probability.

The confusion region, highlighted as the shaded area, is smaller for the Laplace distribution. This indicates that, during contrastive learning, the difference between positive and negative samples is greater, making it more suitable for training. Furthermore, the yellow pentagrams representing the intersection points of both distributions reveal that the Laplace distribution exhibits steeper gradients, signifying a greater impact of randomness.

## B. Dataset and Details

In this paper, we adopt four dataset including colorectal cancer pathological image dataset CRCD (Ye et al., 2023), melanoma pathological image dataset PUMA (Schuiveling et al., 2024), breast cancer pathology image dataset BCSS (Amgad et al., 2019) and multi-class cellular segmentation dataset PanNuke (Gamper et al., 2020). Detailed sample number and category number information about these four datasets is presented in Table 7.

*Table 7.* Dataset details. '#image' stands for number of images, '#tissue' denotes the annotation number of tissues, '#cell' represents annotated cells, '#category' represents category number.

| Dataset | #image | #tissue / #category | #cell / #category |
|---|---|---|---|
| CRCD (Ye et al., 2023) | 1764 | 800 / 9 | 964 / 3 |
| PUMA (Schuiveling et al., 2024) | 151 | 151 / 6 | 151 / 10 |
| BCSS (Amgad et al., 2019) | 151 | 151 / 22 | - |
| PanNuke (Gamper et al., 2020) | 481 | - | 481 / 19 |

*Table 8.* Component Abbreviation Description.

| Abbreviation | Full Name | Abbreviation | Full Name | Abbreviation | Full Name |
|---|---|---|---|---|---|
| ADI | adipose | GLA | glandular secretions | NSM | normal stroma |
| ADR | adrenal | HED | head&neck | OTH | others |
| ANG | angioinvasion | HIT | histiocyte | OUT | outside roi |
| APO | apoptosis | IMM | immune | OVA | ovarian |
| BAC | background | INF | lymphocytic infiltrate | PAN | pancreatic |
| BID | bile duct | KID | kidney | PLA | plasma |
| BLA | bladder | LIV | liver | PRO | prostate |
| BLD | blood | LUN | lung | SKI | skin adnexa |
| BRE | breast | LYM | lymphocyte aggregates | STO | stomach |
| CER | cervix | MEL | melanophage | STR | stroma |
| COL | cloon | MET | metaplasia | SUB | submucosa or serosa |
| DCI | dcis | MUC | mucus | TES | testis |
| END | endothelium | MUS | muscle | THY | thyroid |
| EPI | epidermis | NEC | necrosis | TUM | tumor epithelium |
| ESO | esophagus | NER | nerve | UND | undetermined |
| EXC | exclude | NEU | neutrophil | UTR | uterus |
| FAT | fat | NOR | normal gland | VES | vessel |

## C. Pixel Latent Vector Separation Process Visualization

To more intuitively observe the separation process of pixel latent vectors, we visualized the distribution of pixel latent vectors for each tissue in the PUMA dataset during the training of the diffusion model with contrastive learning. For clarity in visualization, we sampled 50 pixels and adjusted the display quantity of each category based on its proportion relative to the total number of pixels. The separation process of pixel latent vectors is illustrated in Fig. 7. It is evident that as training progresses, pixel latent vectors of different tissues become distinctly separated, while those of the same tissue cluster more closely together.

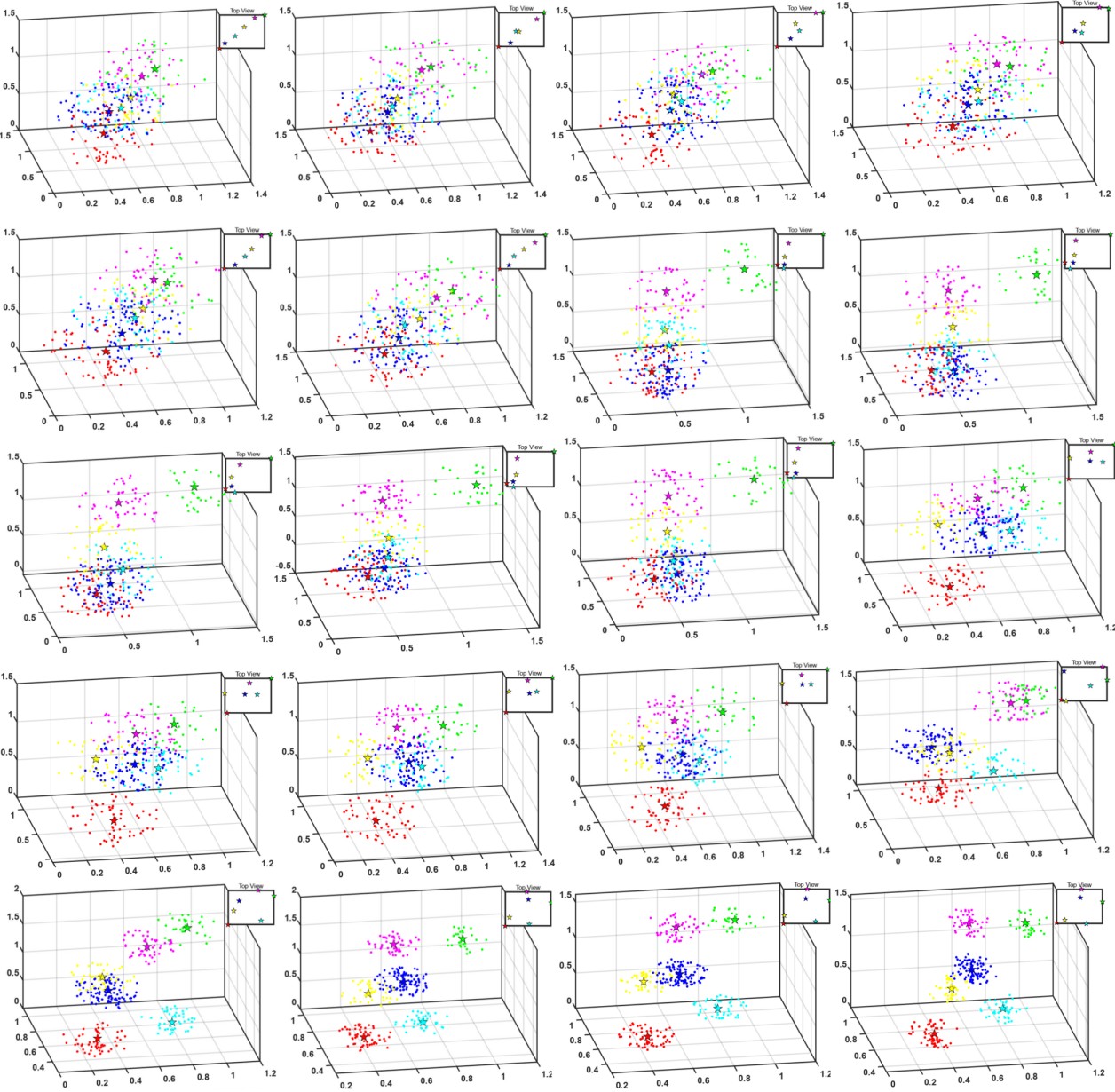

*Figure 7.* Visualization of the pixel latent vector separation process during diffusion model training with contrastive learning. The training spans a total of 100 epochs. After every 5 epochs, the learned pixel latent vectors are reduced to a three-dimensional space using T-SNE(Van der Maaten & Hinton, 2008). Each tissue type is represented by a distinct color.

# D. More Tissue Segmentation Visual Results

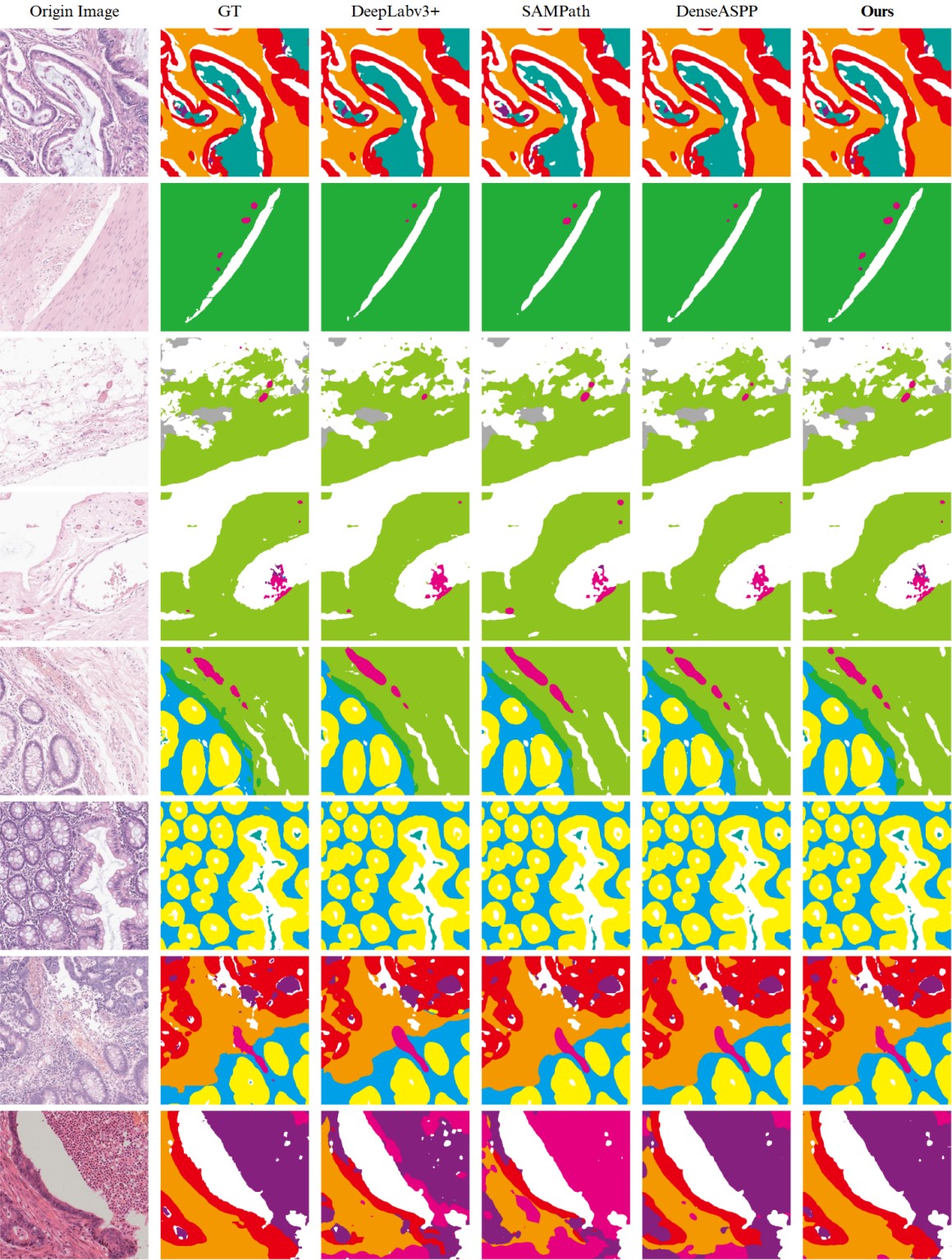

*Figure 8.* More qualitative comparison of tissue segmentation performance. Several leading high-performance methods are selected for visual comparison.

# E. More Cell Segmentation Visual Results

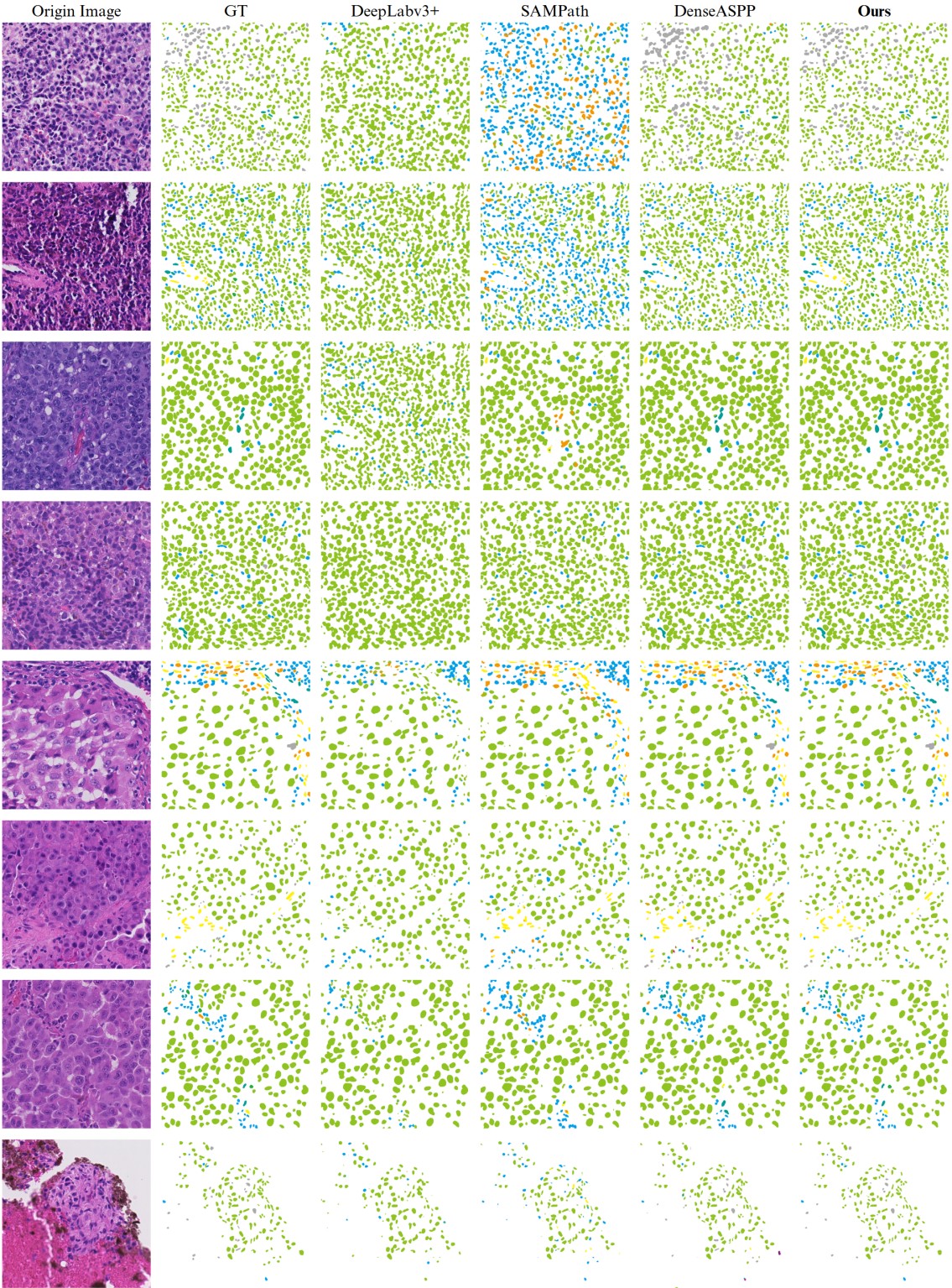

*Figure 9.* More qualitative comparison of cellular segmentation performance. Several leading high-performance methods are selected for visual comparison.

## F. Segmentation Visualization Results on Whole Slide Images

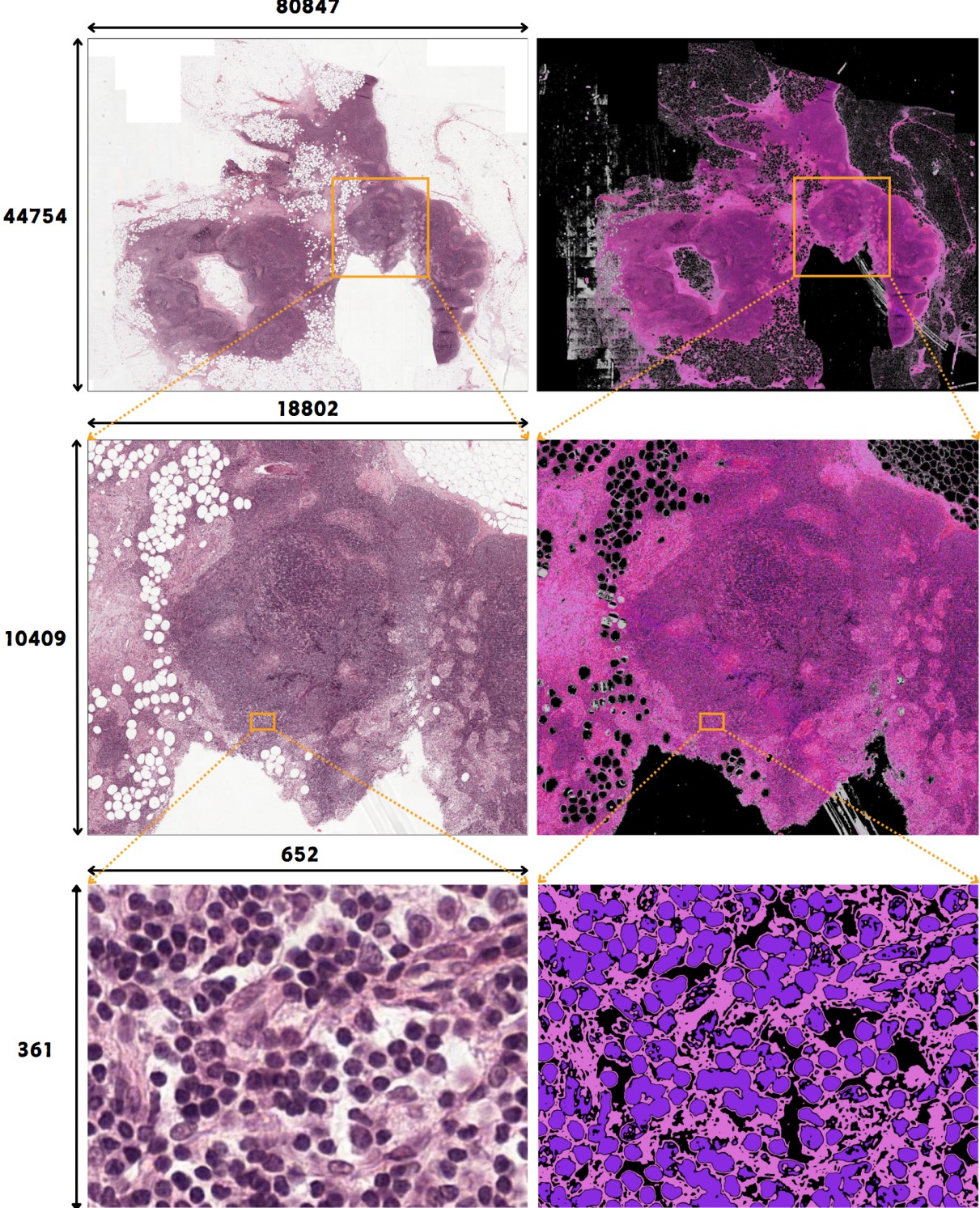

*Figure 10.* The segmentation visualization results of L-Diffusion on pathology whole slide images.

## G. Segmentation Visualization Results on Remote Sensing Images

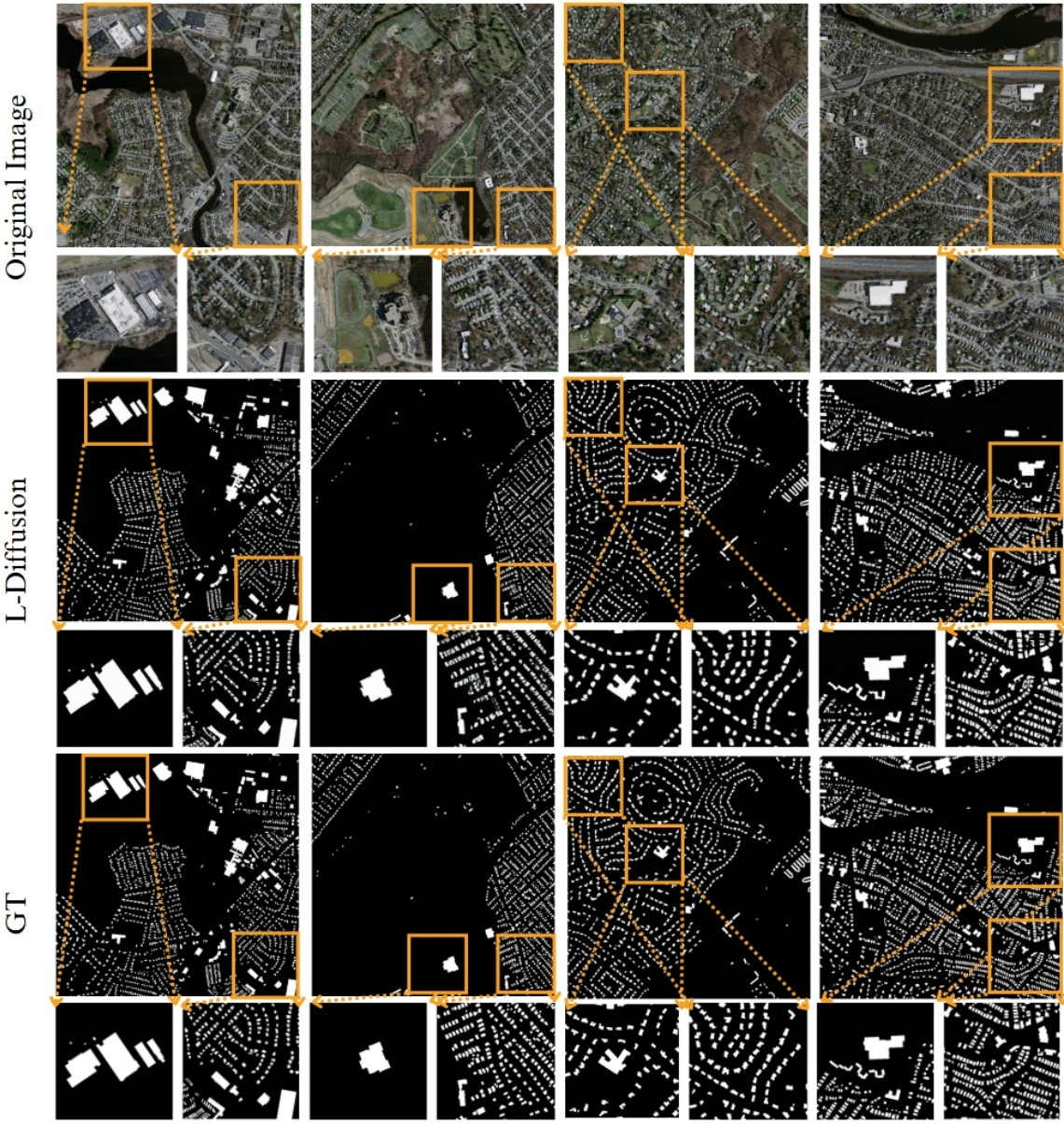

*Figure 11.* The segmentation visualization results of L-Diffusion on remote sensing images.