# OpenReview forum: "L-Diffusion: Laplace Diffusion for Efficient Pathology Image Segmentation"
_ICML.cc/2025/Conference — ICML 2025 poster_

### Official Review · Reviewer_ezdp · 2025-03-11

**Overall Recommendation:** 3

**Summary:**

The paper introduces L-Diffusion, a novel approach to pathology image segmentation by leveraging Laplace distributions and contrastive learning, achieving good performance, and demonstrating generalization capabilities.

**Claims And Evidence:**

Supported Claims:

1) $\textbf{Laplace Distributions Improve Segmentation.}$ Laplace distributions are more effective than Gaussian distributions for modeling pathology image components, especially for tail categories. The authors provide a detailed theoretical analysis comparing the gradients of Laplace and Gaussian distributions, showing that Laplace distributions have sharper gradients and better separation between categories. Empirical results, including visual comparisons of pixel value distributions (Figure 2) and quantitative improvements in segmentation metrics (Tables 1 and 2), further support this claim.

2) $\textbf{Contrastive Learning Enhances Component Separation.}$ The proposed pixel latent vector contrastive learning mechanism improves the separation of different tissue and cellular components. The ablation studies (Table 4) demonstrate that combining contrastive learning with Laplace distributions significantly improves segmentation performance. The visualization of pixel latent vector separation (Figure 7) also provides qualitative evidence of the effectiveness of contrastive learning.

3) $\textbf{Superior Performance on Pathology Image Segmentation.}$ L-Diffusion achieves state-of-the-art performance on multiple pathology image segmentation benchmarks. The paper presents extensive quantitative results (Tables 1 and 2) showing that L-Diffusion outperforms existing methods on datasets such as CRCD, PUMA, BCSS, and PanNuke. The improvements in metrics like DICE, MPA, mIoU, and FwIoU are substantial and well-documented.

4) $\textbf{Generalization to Remote Sensing Images.}$ L-Diffusion generalizes well to other large-scale image segmentation tasks, such as remote sensing. The authors provide quantitative results (Table 5) and qualitative visualizations (Figure 11) showing that L-Diffusion performs competitively on the Massachusetts-Building dataset, a remote sensing image segmentation task.

Claims That Could Benefit from Additional Evidence:

1) $\textbf{Efficiency of L-Diffusion.}$ L-Diffusion is efficient and reduces the dependency on annotated data. While the paper mentions that L-Diffusion can achieve competitive performance with limited annotated data (Table 3), it does not provide a detailed analysis of the computational cost or training time compared to other methods. Diffusion models are generally computationally expensive, and it would be valuable to understand how L-Diffusion compares in terms of resource requirements. Additionally, a sensitivity analysis on the number of diffusion steps (T) and its impact on performance and computational cost would strengthen this claim.

2) $\textbf{Applicability to Other Medical Imaging Modalities.}$ L-Diffusion is broadly applicable to medical image segmentation tasks. The paper focuses on pathology and remote sensing images but does not explore the model's performance on other medical imaging modalities, such as MRI or CT. Extending the experiments to these domains would provide stronger evidence for the model's generalizability.

3) $\textbf{Robustness to Noisy or Imperfect Annotations.}$ L-Diffusion is robust and can handle long-tail distributions effectively. While the paper demonstrates strong performance on tail components, it does not explicitly test the model's robustness to noisy or imperfect annotations, which are common in medical imaging. Including experiments with noisy labels or partial annotations would further validate the model's robustness.

4) $\textbf{Ethical and Clinical Impact.}$ L-Diffusion provides a powerful tool for advancing tumor diagnosis and microenvironment analysis. The paper briefly mentions ethical approval for the datasets but does not discuss the broader ethical implications or clinical impact of using L-Diffusion in real-world medical settings. A more detailed discussion of potential risks, limitations, and guidelines for deployment would strengthen this claim.

**Essential References Not Discussed:**

I think the Laplace noise was previously found to be better schedule for image generation. (See https://arxiv.org/abs/2407.03297  and https://arxiv.org/abs/2304.05907)

**Experimental Designs Or Analyses:**

The authors evaluate L-Diffusion on several well-established pathology image datasets, including CRCD (colorectal cancer), PUMA (melanoma), BCSS (breast cancer), and PanNuke (multi-class cellular segmentation). These datasets are representative of the challenges in pathology image segmentation, such as gigapixel resolution, multi-scale features, and long-tail distributions.

The paper uses standard evaluation metrics for segmentation tasks, including DICE, MPA (Mean Pixel Accuracy), mIoU (Mean Intersection over Union), and FwIoU (Frequency Weighted IoU). These metrics are widely accepted in the medical imaging community and provide a balanced assessment of segmentation performance.

The authors compare L-Diffusion with a variety of state-of-the-art methods, including U-Net++, Swin-UNet, DeepLabv3, and SAMPath.

The paper does not provide a detailed analysis of the computational cost or training time compared to other methods. Diffusion models are generally computationally expensive, and it would be valuable to understand how L-Diffusion compares in terms of resource requirements.

The paper does not discuss the sensitivity of L-Diffusion to key hyperparameters, such as the scale parameter (b) in the Laplace distribution.

**Methods And Evaluation Criteria:**

Yes, the proposed methods and evaluation criteria in the paper are well-suited for the problem of pathology image segmentation and the broader application of medical image analysis.

**Other Comments Or Suggestions:**

not found

**Other Strengths And Weaknesses:**

see above

**Questions For Authors:**

While the paper focuses on Laplace distributions, have the authors explored other distributions, such as Cauchy or Student's t, for modeling pathology image components? If so, how do these distributions compare to Laplace distributions in terms of segmentation performance?

How does L-Diffusion compare to unsupervised or semi-supervised methods for pathology image segmentation, particularly in scenarios with limited annotated data? Have the authors explored unsupervised or semi-supervised variants of L-Diffusion?

**Relation To Broader Scientific Literature:**

Medical image segmentation, particularly in pathology, has been extensively studied using deep learning models such as U-Net (Ronneberger et al., 2015), DeepLab (Chen et al., 2017), and more recently, Transformers (Atabansi et al., 2023). These models have shown success in segmenting tissues and cells in pathology images, but they often struggle with long-tail distributions and multi-scale features. L-Diffusion addresses these challenges by introducing Laplace distributions and contrastive learning to enhance the separation of different components, particularly tail categories. This builds on prior work by providing a novel approach to handling the inherent complexities of pathology images, such as gigapixel resolution and imbalanced tissue distributions.

Diffusion models, such as Denoising Diffusion Probabilistic Models (DDPM) (Ho et al., 2020), have gained popularity for their ability to model complex data distributions. These models have been applied to various tasks, including image generation, denoising, and segmentation. However, most prior work in diffusion models uses Gaussian distributions for noise modeling. L-Diffusion introduces Laplace distributions as an alternative to Gaussian distributions in diffusion models. The authors argue that Laplace distributions provide sharper gradients and better separation between different categories, making them more suitable for pathology image segmentation. This is a novel contribution that extends the applicability of diffusion models to medical imaging tasks with long-tail distributions.

Contrastive learning has emerged as a powerful technique in self-supervised learning, particularly in computer vision (Chen et al., 2020). It has been applied to various tasks, including image classification, object detection, and segmentation. In medical imaging, contrastive learning has been used to improve feature representations and reduce the dependency on annotated data. L-Diffusion incorporates pixel latent vector contrastive learning to enhance the separation of different tissue and cellular components. This builds on prior work by applying contrastive learning to the latent space of diffusion models, which is a novel approach. The authors demonstrate that contrastive learning significantly improves segmentation performance, particularly for tail components, by amplifying the distributional differences between different categories.

**Theoretical Claims:**

The theoretical claims and proofs in the paper are generally correct and well-supported by mathematical derivations. The authors effectively demonstrate why Laplace distributions are more suitable than Gaussian distributions for pathology image segmentation, and they provide a solid theoretical foundation for the proposed L-Diffusion model.

---

> ### Author Rebuttal · Authors · 2025-04-01
>
> ## Reply to Reviewer ezdp
>
> We sincerely appreciate your efforts in reviewing our paper and for your constructive feedback. We have organized your comments and provided our responses below, hoping they address your concerns.
>
> **[Question (Q)1] Efficiency of L-Diffusion**
>
> Answer (A)1: Thanks. L-Diffusion consists of two stages: Latent feature extraction with diffusion model and segmentation with ConvNeXT. The former is time-consuming.
>
> In the training stage, employing two A6000 GPU resources and a dataset of 50,000 patch samples, the L-Diffusion model requires about 26 hours—24 hours for training the diffusion model and 2 hours for refining the ConvNeXT segmenter. Meanwhile, SOTA SAMPath achieves convergence within 5 hours. Notably, L-Diffusion's two-stage design enhances ConvNeXT fine-tuning efficiency.
>
> In the inference stage, L-Diffusion and SAMPath take an average of 2.5 and 1.8 hours per image of size 90027 × 88341, respectively. For more results, see **A2 to reviewer ZaXb**. Future work will focus on accelerating L-Diffusion.
>
> **[Q2] Applicability to Other Medical Imaging Modalities**
>
> A2: Thanks. We have added the generalization experiment. Results show that L-Diffusion achieves 6.43% and 0.67%  improvements to SOTA methods on BraTS and RIM-ONE datasets, respectively. Please refer to **A1 to reviewer ZaXb** for detailed experiment results. We will supplement the final version with more analysis and visualizations.
>
> **[Q3] Robustness to Noisy or Partial Annotations**
>
> A3: Thanks. Table 3 of the original submission shows the experiment with partial annotations. The pathology usually contains space-noise (wrong boundary) and label-noise (wrong type). The noise robustness experiments (DICE score) are given as:
> |Type|5%|10%|20%|30%|
> |-|-|-|-|-|
> |space-noise|92.18|91.83|91.79|91.65|
> |label-noise|88.72|85.26|80.14|75.48|
>
> Our method is robust to space-noise due to only the inner high-confidence area pixel is adopted for contrastive learning. Similar to other methods, our method is also sensitive to label-noise, which can be solved by the anti-noise learning loss function. More results and analysis will be included in the final version.
>
> **[Q4] Ethical and Clinical Impact Discussion**
>
> A4: Thanks. We recognize the importance of addressing the ethical and clinical implications of deploying L-Diffusion in real-world medical applications. While we briefly mention ethical approvals for datasets, we will expand our discussion to cover potential risks, such as biases in dataset annotations, the need for regulatory validation before clinical use, and challenges in interpretability.
>
> **[Q5] Sensitivity of Key Hyperparameters**
>
> A5: Sorry for the confusion. The scale parameter (red box of Eqn. 15) is an adaptive parameter predicted by U-Net. Ablation studies on different distribution and diffusion steps are given in Table 4 and Figure 5. Moreover, an ablation study on τ of contrastive learning is given as follows:
> |τ|0.1|0.08|0.05|0.02|
> |-|-|-|-|-|
> |DICE|89.33|90.67|92.11|88.74|
>
> We will add it to the final version.
>
> **[Q6] (a) Relation to Prior Work ([1] Imp... [2] Diff...) & (b) Exploration of Alternative Distributions (Dis.)**
>
> A6: Thank you for your comments.
>
> **(a)**: *[1]* demonstrates that the Laplace Dis. has steeper peaks and heavier tails. This property makes it more advantageous when dealing with data with outliers or sparse noise. *[2]* compared the Gaussian Dis., t Dis., Uniform Dis., and so on tend to produce smooth samples for image generation tasks.
>
> Different from *[1,2]*, this paper proposes for the first time to achieve segmentation through sharp dis. alienation latent space and gives a detailed theoretical derivation.
>
> We will add the above papers to the related work and highlight our contributions.
>
> **(b)**: We add the comparative experiment as follows:
> |Dis.|DICE|Runtime|
> |-|-|-|
> |Cauchy|16.25|7325|
> |Student's T|83.17|20440|
> |Laplace|85.75|8882|
>
> In summary, Laplace Dis. maintains its superiority. This is because the Cauchy dis. has no expectations and variances, making gradient optimization less stable. Student's t Dis. has potential in accuracy, but its runtime is too long because of its complex gradient solution. We are committed to adding detailed data and analysis to the final manuscript.
>
> **[Q7] Exploration of Unsupervised or Semi-supervised**
>
> A7: We appreciate the reviewer's interest in the applicability of L-Diffusion to unsupervised and semi-supervised variants. Since L-Diffusion is based on contrast learning in the diffusion stage, this model supports a semi-supervised variant. We explored semi-supervised effects in Table 3.
>
> For the unsupervised variant, only the latent feature is extracted by the L-diffusion, and then the clustering algorithm is adopted K-Means for the unsupervised method. The performance on the RIM-ONE dataset is provided below:
> |Type|Full.|Semi.|Un.|
> |-|-|-|-|
> |DICE|96.12|88.91|60.92|
>
> More details and analysis will be given in the final version.

---

> > ### Comment · Reviewer_ezdp · 2025-04-03
> >
> > Thanks for the authors, I am raising my score to 3

---

### Official Review · Reviewer_9Pf9 · 2025-03-12

**Overall Recommendation:** 4

**Summary:**

This paper introduces L-Diffusion, an innovative framework designed to advance pathology image segmentation by utilizing Laplace distributions and contrastive learning.  The primary contribution of the paper lies in its use of Laplace distributions to model distinct components within pathology images, which enhances distributional divergence and facilitates more precise and robust segmentation.
The novel pixel latent vector contrastive learning mechanism further reduces reliance on annotated data, addressing the challenges associated with long-tail components.  The approach significantly improves segmentation performance on tissue and cell datasets, showing substantial gains over existing methods.

**Claims And Evidence:**

Yes, the claims made in the submission are generally supported by clear and convincing evidence.   The authors provide both theoretical analysis and extensive experimental evaluations to substantiate their claims about the effectiveness of the L-Diffusion framework for pathology image segmentation.

- Theoretical Analysis:
The paper clearly explains the rationale behind using Laplace distributions for component modeling, as opposed to Gaussian distributions.   It presents mathematical derivations comparing the gradients of the two distributions, showing that Laplace distributions are more sensitive to noise, which is beneficial for pathology image segmentation tasks.

- Experimental Evidence:
The paper includes quantitative results demonstrating the improvements in segmentation performance across various tissue and cell datasets (CRCD, PUMA, BCSS, PanNuke) compared to several state-of-the-art models.   The reported results are statistically significant, showing substantial performance gains (e.g., improvements in DICE, MPA, mIoU, and FwIoU metrics).
The qualitative visualizations further support these findings, showing that L-Diffusion achieves better boundary segmentation and handles tail-class components more effectively.

- Comparison to Existing Methods:
The authors compare L-Diffusion with several mainstream segmentation models (e.g., U-Net++, DeepLab, Swin-UNet) and show that their model consistently outperforms these methods, especially in terms of segmenting components with lower proportions (long-tail distribution).

- Ablation Studies:
The paper conducts detailed ablation studies to show the importance of each component of L-Diffusion, particularly the integration of the Laplace distribution and contrastive learning, which provides additional evidence.

**Essential References Not Discussed:**

In my personal opinion, this paper provides a thorough introduction to the algorithms and ideas involved in the relevant work section and experimental implementation. The approach is innovative, and there are no instances of withholding key literature that would be crucial to the paper's significance.

**Experimental Designs Or Analyses:**

Yes, the experimental designs and analyses appear sound and provide strong evidence for the claims made in the paper.

1. Benchmark Datasets:
The paper uses a variety of benchmark datasets (e.g., CRCD, PUMA, BCSS, and PanNuke) for testing the model. These datasets cover different aspects of pathology image segmentation, including tissue and cell segmentation, and represent diverse challenges such as multi-scale features and long-tail distributions. The choice of datasets is appropriate for testing the model's generalizability across multiple types of pathology image segmentation tasks.

2. Comparison with SOTAs:
The paper compare L-Diffusion with a variety of SOTA methods (e.g., U-Net, DeepLabv3, Swin-UNet, FastFCN), which is a solid approach to demonstrate the advantages of their proposed method. The reported quantitative results show significant improvements across tissue and cell segmentation datasets, providing strong evidence of the method’s effectiveness.

3. Qualitative and Quantitative Results:
The qualitative results (visualizations) demonstrate that L-Diffusion performs well in segmenting boundaries and handling tail-class components, which aligns with the claims made by the authors. The quantitative results (based on the metrics mentioned) show clear improvements over existing methods, further supporting the paper’s claims of better performance.

4. Ablation Studies:
The paper conduct ablation studies to isolate the impact of key components in their model, such as the integration of Laplace distributions and contrastive learning. This is an essential and well-designed experiment, as it provides insights into which parts of the model contribute most to its success. The ablation results confirm that the combination of these two techniques is a major factor in the model's improved performance.

**Methods And Evaluation Criteria:**

Yes, the proposed methods and evaluation criteria are well-suited for pathology image segmentation.  Using Laplace distributions to model distinct components effectively addresses multi-scale features and long-tail distributions in pathology images, enhancing segmentation precision, especially for rare components.  The integration of contrastive learning further refines component differentiation while reducing reliance on annotated data.  The use of diffusion steps to refine feature maps and latent vectors contributes to more accurate segmentation.

The evaluation metrics (DICE, MPA, mIoU, and FWIoU) are standard and effective, assessing various aspects of segmentation performance, including accuracy, precision, and class imbalance.  Qualitative visualizations complement these metrics by showing better boundary segmentation, particularly for tail-class components.

The benchmark datasets, such as CRCD, PUMA, BCSS, and PanNuke, cover diverse pathology segmentation tasks, while the inclusion of a remote sensing dataset for generalization tests demonstrates the method's robustness across different domains.  Overall, the proposed methods and evaluation criteria are appropriate and robust for addressing the challenges in pathology image segmentation.

**Other Comments Or Suggestions:**

I personally suggest that the Related Work section be placed after the Introduction to help readers quickly engage with the core implementation ideas.At the same time, the specific meaning of the abbreviation is given in the dataset section of the appendix, so that the reader can better understand the processing effect of the segmentation model on different categories.

**Other Strengths And Weaknesses:**

As mentioned earlier, this paper overall meets the standards for publication at ICML. I believe the authors' L-Diffusion approach also offers valuable insights for feature engineering implementation.

**Questions For Authors:**

N/A

**Relation To Broader Scientific Literature:**

The contributions of this paper are built upon a broad foundation of existing research, spanning multiple areas such as pathology image segmentation, diffusion models, and contrastive learning. The practical application of L-Diffusion is positioned within the realm of pathology image segmentation, following the Diffusion + Pathology paradigm. Interestingly, the authors have innovatively approached the problem by utilizing component distributions across diffusion steps, which not only fine-tunes the decomposition of pathological semantic information but also enhances the richness of the data through the diffusion process. This represents a significant innovation in addressing pathology image segmentation challenges. Furthermore, the introduction of contrastive learning aligns with ideas from multimodal learning and other related fields.

Overall, the L-Diffusion model presented in this paper is convincing in its relation to the broader scientific literature, adhering to sound research principles. The novel application of the Laplace diffusion process adds a substantial innovation within the field, making the approach a noteworthy contribution to the domain.

**Theoretical Claims:**

Yes, I focused on the rationale behind applying Laplace distributions to pathology image segmentation, particularly in terms of the formulaic principles. This is thoroughly addressed in Section A, "Mathematical Derivations", in the appendix, and the visualization of the differences between the Laplace and Gaussian distributions is commendable. It provides convenience for readers when interpreting the paper. The formal proof and the visualizations in the paper demonstrate that the probability change range $\Delta y$ of the Laplace distribution is larger than that of the Gaussian distribution $\Delta y'$, which aligns well with the segmentation strategy for pathology images. Additionally, the introduction of contrastive learning, which brings similar class samples closer and pushes different class samples apart, is a sound approach both conceptually and methodologically.

---

> ### Author Rebuttal · Authors · 2025-04-01
>
> ## Reply to Reviewer 9Pf9
>
> We sincerely appreciate your thorough and insightful review of our paper, L-Diffusion: Laplace-Based Diffusion Model for Pathology Image Segmentation. Your positive evaluation of our contributions, including the use of Laplace distributions, contrastive learning, and the overall experimental design, is highly encouraging. We address your comments and suggestions below:
>
> **[Question (Q)1] Reordering the Related Work Section**
>
> Thank you for your constructive comments. We acknowledge your suggestion to move the Related Work section after the Introduction to enhance readability. In the revised version, we will adjust the structure accordingly.
>
> **[Q2] Clarification of Dataset Abbreviations**
>
> Thank you for your valuable suggestions. We agree that explicitly defining dataset abbreviations in the appendix would enhance clarity. We will ensure that all dataset names and category labels are fully defined to improve readability for the audience.

---

### Official Review · Reviewer_nPmf · 2025-03-12

**Overall Recommendation:** 4

**Summary:**

This paper proposes a new diffusion-based method to tackle pathology image segmentation. The pathology image segmentation is a challenging task because of the large, gigapixel resolution, diverse scales, and imbalanced tissue distributions in these images.  Traditional segmentation models like U-Net and DeepLab have shown promise but struggle with labor-intensive annotation and feature extraction for tail categories. To address these issues, the paper introduces the Laplace Diffusion Model (L-Diffusion), which leverages Laplace distributions instead of Gaussian to model distinct components within the images. L-Diffusion uses contrastive learning to enhance the differentiation between components while maintaining intra-component similarity, improving segmentation precision and robustness, particularly for tail components. The model reduces the reliance on annotated data by capturing the distributional characteristics of different components. Extensive experiments show that L-Diffusion outperforms existing methods in accuracy and robustness across various benchmarks, providing an innovative approach to pathology image segmentation.

**Claims And Evidence:**

Based on the challenges summarized in the submission: "current pathology image segmentation tasks grapple with labor-intensive annotation processes or limited accuracy in identifying tail samples", there are two main claims made by the submission:
(1). Laplace distribution is advantageous for broadening distribution disparities based on the analysis in the submission.
(2). Laplace diffuse model learns better component distribution  for pathology image segmentation, due to it's long-tail nature.
These two claims are well supported by the method theory and experiments.

**Essential References Not Discussed:**

N.A.

**Experimental Designs Or Analyses:**

There are 3 main experiments conducted for evaluating the effectiveness of the proposed method: (1) Quantitative evaluations agains baseline methods (2) Ablation study (3) Generalization on large-scale data.

The quantitative evaluations are made comprehensively with many state of the art methods and various metrics. The proposed method seems to get a decent gain consistently on each of the benchmarking dataset.

**Methods And Evaluation Criteria:**

The proposed method has been evaluated in the context of pathology image segmentation on a series of benchmarking datasets: CRCD , PUMA, BCSS, against a series of state of the art baseline methods

**Other Comments Or Suggestions:**

1. The equations in the paper section 3 is not numbered, making it difficult for reference.

**Other Strengths And Weaknesses:**

Strengths
1. The reasoning in Section 3.1 is sound in term of showing the comparison between Laplace distribution gradients and Guassian distribution gradients.
2. The experiments are comprehensive and the code is available anonymously. The experiment is compare a large number of baseline methods and the proposed method is outperforming all the previous methods consistently.

**Questions For Authors:**

The authors are suggested to address the concerns listed in the previous sections during rebuttal period.

**Relation To Broader Scientific Literature:**

The proposed Laplace model can be applied to a broader domains such as image segmentation e-commerce product data, which also shows a long-tail distribution.

**Theoretical Claims:**

1. The proofs in Section 3.1 for computing the gradient of Laplace distribution is correct.
2. The methmetical dereivation in supplementary material A. is verified.

---

> ### Author Rebuttal · Authors · 2025-04-01
>
> ## Reply to Reviewer nPmf
>
> We appreciate the reviewers' detailed and insightful feedback on our work. Below, we address the key points raised in the review.
>
>
> **Response to Theoretical Claims**
>
> We are grateful for the acknowledgment that our theoretical derivations and mathematical proofs are correct. We appreciate the reviewers' verification of our gradient computations for the Laplace distribution in Section 3.1 and the mathematical derivations in Supplementary Material A.
>
> **Response to Experimental Design and Evaluation**
>
> We thank the reviewers for their recognition of the comprehensive nature of our experiments, including:
>
> - Quantitative evaluation against strong baseline methods.
>
> - Ablation studies that highlight the contributions of different components of our method.
>
> - Generalization experiments on large-scale datasets, demonstrating the robustness of our approach.
>
> The consistent performance improvements across benchmarking datasets (CRCD, PUMA, BCSS) further validate the effectiveness of L-Diffusion.
>
>
> **Response to Weaknesses and Suggestions**
>
> Thank you for your valuable suggestion. We acknowledge the reviewer's concern that the equations in Section 3 are not numbered, which makes it difficult for reference. We will add equation numbers in the final version to improve clarity and readability.

---

### Official Review · Reviewer_ZaXb · 2025-03-13

**Overall Recommendation:** 4

**Summary:**

The paper introduces L-Diffusion, a novel Laplace Diffusion Model designed for efficient pathology image segmentation. Unlike traditional approaches relying on Gaussian distributions, L-Diffusion employs multiple Laplace distributions to better differentiate component features in pathology images. The model follows a diffusion process, generating a sequence of feature maps, and enhances pixel-wise vector representations using contrastive learning. This approach significantly improves segmentation performance, particularly for tail-class components, which are often difficult to identify due to the long-tail distribution in pathology images. Extensive experiments on six tissue and cell segmentation datasets show that L-Diffusion achieves substantial improvements over state-of-the-art models, with up to 7.16% higher DICE score for tissue segmentation and 20.09% improvement for cell segmentation. The paper also provides theoretical analysis supporting the advantages of using Laplace distributions, demonstrating their ability to enhance component differentiation and model efficiency.

**Claims And Evidence:**

The claims made in the paper are well-supported by both theoretical analysis and experimental results. The authors claim that using Laplace distributions instead of Gaussian distributions enhances feature decomposition and improves segmentation accuracy, particularly for tail-class components. This claim is backed by mathematical derivations and empirical comparisons of Gaussian vs. Laplace distribution differentiation, which show that the latter leads to greater separability of pixel-wise feature vectors.

**Essential References Not Discussed:**

The paper provides a comprehensive discussion of related literature, covering key works in diffusion models, contrastive learning, and pathology image segmentation. It cites foundational methods such as U-Net, DeepLab, and Transformer-based models, as well as recent diffusion-based segmentation approaches like MedSegDiff. The discussion on Laplace distributions and their role in enhancing feature separability is well-supported by prior probabilistic modeling research. Additionally, the application of contrastive learning to pathology image segmentation is contextualized within the broader field of self-supervised learning. Overall, the references are sufficient, and no essential prior works appear to be missing.

**Experimental Designs Or Analyses:**

The experimental design is generally sound and appropriate for pathology image segmentation. The model is tested on six diverse datasets, covering both tissue and cell segmentation, ensuring broad applicability. Metrics such as DICE, MPA, mIoU, and FwIoU are correctly chosen for evaluating segmentation performance. The comparison with state-of-the-art models is comprehensive, showing clear performance improvements. Ablation studies confirm the contributions of Laplace distributions and contrastive learning, and a generalization test on a remote sensing dataset suggests potential broader applicability. A minor limitation is the lack of evaluation on other medical imaging domains, which could further validate the model’s versatility.

**Methods And Evaluation Criteria:**

The proposed L-Diffusion method and evaluation criteria are well-suited for pathology image segmentation. The use of Laplace distributions enhances feature differentiation, and contrastive learning improves pixel-wise representation, addressing challenges in long-tail class segmentation. The model is evaluated on six benchmark datasets covering both tissue and cell segmentation, ensuring a comprehensive assessment. Metrics such as DICE, MPA, mIoU, and FwIoU are appropriate for measuring segmentation accuracy and robustness.

**Other Comments Or Suggestions:**

The second sentence in the third paragraph of the Introduction seems a bit unclear.

**Other Strengths And Weaknesses:**

### Strengths

1. The paper introduces a novel approach by replacing Gaussian distributions with Laplace distributions, improving feature separability for pathology image segmentation.
2. The use of pixel latent vector contrastive learning enhances segmentation accuracy, especially for tail-class components, addressing long-tail distribution challenges.
3. L-Diffusion outperforms state-of-the-art segmentation models on six benchmark datasets, demonstrating significant improvements in both tissue and cell segmentation.
4. The mathematical derivations provide a solid foundation for Laplace-based diffusion modeling, supporting the claimed improvements in feature differentiation.

### Weaknesses

1. While the model performs well on pathology images, how would it generalize to other medical imaging tasks such as radiology or ophthalmology?
2. The proposed diffusion process involves multiple steps; can the authors provide runtime comparisons with other state-of-the-art segmentation models to assess efficiency?
3. The model introduces multiple hyperparameters (e.g., diffusion steps, contrastive learning settings). How sensitive is the performance to these hyperparameters, and could the authors provide guidelines for optimal tuning?

**Questions For Authors:**

Please see the weaknesses part above.

**Relation To Broader Scientific Literature:**

The paper builds on diffusion models and contrastive learning, extending them to pathology image segmentation. Traditional segmentation models, such as U-Net, DeepLab, and Transformers, have been widely used, but they struggle with long-tail class segmentation and multi-scale feature extraction. The introduction of Laplace distributions instead of Gaussian distributions aligns with prior work on improving feature separability in latent spaces. Contrastive learning, which has been effective in self-supervised learning, is adapted here to enhance pixel-wise feature differentiation. The paper also connects to recent efforts in medical image segmentation using diffusion models, such as MedSegDiff [1], but uniquely applies component-wise latent distribution modeling.

[1] Wu, Junde, et al. MedSegDiff: Medical image segmentation with diffusion probabilistic model.

**Theoretical Claims:**

The paper provides theoretical justification for using Laplace distributions over Gaussian distributions, arguing that the steeper gradient of the Laplace distribution enhances feature differentiation. The derivations, including probability density functions, gradient calculations, and diffusion step equations, appear mathematically sound. The Laplace noise formulation and reverse diffusion process are derived systematically, and the transition from Gaussian to Laplace-based modeling is well-supported. While I did not rigorously verify every step, the overall framework aligns with established diffusion model theory. No obvious errors were found, but external validation would further confirm the proofs' correctness.

---

> ### Author Rebuttal · Authors · 2025-04-01
>
> ## Reply to Reviewer ZaXb
>
> We sincerely appreciate the reviewer's valuable feedback and insightful comments on our paper. We are pleased that the reviewers recognize our contributions in introducing L-Diffusion, leveraging Laplace distributions for pathology image segmentation, and improving segmentation performance, particularly for tail-class components. Below, we address the key concerns and provide clarifications on generalization, efficiency, and hyperparameter sensitivity.
>
> **[Question (Q)1] How would it generalize to other medical imaging tasks such as radiology or ophthalmology?**
>
> Answer (A)1: Thanks to the reviewers for their interest in the wider applicability of L-Diffusion. While our work focuses on pathology image segmentation, the principles of Laplace-based diffusion modeling and pixel latent vector contrastive learning are broadly applicable. Many medical imaging tasks, such as radiology (CT, MRI) and ophthalmology (fundus images, OCT), share challenges like heterogeneous textures, fine-grained structures, and class imbalances, suggesting that L-Diffusion could generalize well to these domains.
>
> We have conducted preliminary experiments on a public brain tumor MRI dataset and the RIM-ONE glaucoma dataset. On the BraTS dataset, our DICE score is better than **SOTA (83.13) [1]**. On the RIM-ONE glaucoma dataset, our MPA score is better than **SOTA (95.45) [2]**. Our findings indicate that L-Diffusion retains its advantages in segmenting small, rare tumor components. We are committed to adding statistics and visualizations to the revised manuscript.
>
> | DATASET | DICE | MPA | mIoU | FwIoU |
> |-|-|-|-|-|
> | BraTS | 89.56 | 84.53 | 86.41 | 87.10 |
> | RIM-ONE | 96.12 | 95.58 | 94.39 | 95.27 |
>
> **[Q2] Can the authors provide runtime comparisons with other state-of-the-art segmentation models to assess efficiency?**
>
> A2: Thank you for the constructive suggestion.
> - Training phase: Since our method is based on stable diffusion backbone, core training runtime is only in the ConvNeXT classification header, which is more advantageous than other full-training models.
> - Inference phase: The main use scenario is gigapixel images, which means that for these images, the actual image processing time is much longer than the segmentation time. This means that we can process tasks and segmentation tasks in parallel to reduce the runtime disadvantages of the model. We provide a table to compare the efficiency of different models so that you can more intuitively realize the impact of runtime disadvantage on efficiency.
>
> | Method | Image Size | Processing Time | Segmentation Time | Total Time |
> |:-:|:-:|:-:|:-:|:-:|
> | DeepLabV3+ | 90027 × 88341 | 763.58s | 382.85s | 763.63s |
> | SAMPath | 90027 × 88341 | 763.58s | 6355.31s | 6355.31s |
> | L-Diffusion | 90027 × 88341 | 8880.96s | 4899.84s | 8881.60s |
>
> In summary, L-Diffusion has advantages over traditional models in the training phase and is about 30% slower than SOTA in the inference phase, which is acceptable at runtime. In future work, we will consider introducing a quantitative accelerated diffusion model and other means for optimization, and the above data and analysis will be supplemented to revise the manuscript.
>
> **[Q3] How sensitive is the performance to these hyperparameters, and could the authors provide guidelines for optimal tuning?**
>
> A3: We greatly appreciate the reviewer's insightful question regarding hyperparameter sensitivity and tuning.
>
> Hyperparameters mentioned in the original text (**Models and Parameters** in the Experiment):
> - Diffusion steps: Too few steps limit denoising capacity, while too many increase computational cost without substantial improvement. This is reflected in Fig. 5. (5~15)
> - Sampling number of contrastive learning: Similar to the diffusion step. (100)
> - Learning rate of Adam optimizer: Diffusion Training (0.00001), Segmentation Training (0.001)
> - Batch size: Diffusion Training (1), Segmentation Training (32)
>
> Unmentioned hyperparameters:
> - Temperature scaling of contrastive learning: In the case of avoiding gradient explosion, smaller temperature scales can help sharpen the distribution. (0.05-0.1)
>
> An ablation study on temperature scaling τ of contrastive learning is given as follows:
> |τ|0.1|0.08|0.05|0.02|
> |-|-|-|-|-|
> |DICE|89.33|90.67|92.11|88.74|
>
> We will include practical tuning guidelines in the revised manuscript.
>
> **[Q4] The second sentence in the third paragraph of the Introduction seems a bit unclear.**
>
> A4: Thank you very much for your correction. We have checked the original text and provided the revised version.
>
> Revised text: Prominent deep learning models, including U-Net (R~), DeepLab (L~), and Transformer (A~), have demonstrated superior performance across a spectrum of medical image segmentation tasks.
>
> **References**
>
> [1] https://arxiv.org/pdf/2403.09262
>
> [2] https://arxiv.org/pdf/1903.02740

---

### Decision · Program_Chairs · 2025-05-01

**Decision:**

Accept (poster)

**Comment:**

All reviewers appreciated the originality of using Laplace distributions within a diffusion framework, especially for the theoretical proofs which are carefully derived and validated by multiple reviewers (ZaXb, nPmf, 9Pf9, ezdp). This work demonstrates clear advantages over Gaussian distributions through sharper gradients and improved component separability. Empirically, L-Diffusion consistently outperforms state-of-the-art segmentation models across six diverse datasets (e.g., CRCD, PUMA, BCSS, PanNuke), achieving substantial improvements with DICE score gains of up to 7.16% for tissue segmentation and 20.09% for cell segmentation, supported by strong quantitative metrics and compelling qualitative visualizations. All the reviewers gave positive rating to this work and the novelty and contributions of this work have been accepted by all the reviewers. Therefore, meta-reviewer agrees with the recommendation of the reviewers.